# Efficient and Sharp Off-Policy Learning under Unobserved Confounding

**Konstantin Hess[1,2,\*], Dennis Frauen[1,2], Valentyn Melnychuk[1,2], Stefan Feuerriegel[1,2]**

[1]LMU Munich     [2]Munich Center for Machine Learning
[\*]Corresponding author: `k.hess@lmu.de`

## Abstract

We develop a novel method for personalized off-policy learning in scenarios with unobserved confounding. Thereby, we address a key limitation of standard policy learning: standard policy learning assumes unconfoundedness, meaning that no unobserved factors influence both treatment assignment and outcomes. However, this assumption is often violated, because of which standard policy learning produces biased estimates and thus leads to policies that can be harmful. To address this limitation, we employ causal sensitivity analysis and derive a *semi-parametrically efficient estimator for a sharp bound on the value function under unobserved confounding*. Our estimator has three advantages: (1) Unlike existing works, our estimator avoids unstable minimax optimization based on inverse propensity weighted outcomes. (2) Our estimator is semi-parametrically efficient. (3) We prove that our estimator leads to the optimal confounding-robust policy. Finally, we extend our theory to the related task of policy improvement under unobserved confounding, i.e., when a baseline policy such as the standard of care is available. We show in experiments with synthetic and real-world data that our method outperforms simple plug-in approaches and existing baselines. Our method is highly relevant for decision-making where unobserved confounding can be problematic, such as in healthcare and public policy.

## 1 Introduction

Policy learning is crucial in many areas such as healthcare (Feuerriegel et al., 2024; Kraus et al., 2023), education (Chan, 2023), and public policy (Ladi & Tsarouhas, 2020). However, collecting data through randomized experiments is often either infeasible or unethical. Instead, methods are needed that use observational data to inform decision-making. Here, we focus on *off-policy learning* to optimize decision policies from observational data (Athey & Wager, 2021).

The reliability of standard off-policy learning is compromised when *unobserved confounding* is present (Kallus et al., 2019). Unobserved confounding arises when factors affect both treatment choices and outcomes but are not recorded (Pearl, 2009). For example, the race of a patient may – unfortunately – affect the access to treatments (Obermeyer et al., 2019), yet race is typically not recorded in patient records. Hence, standard off-policy learning that relies on the assumption of no unobserved confounding will lead to *biased* estimates and may even generate *harmful* policies.

As a remedy, *confounding-robust policy learning* aims to find the optimal policy under worst-case unobserved confounding. This is typically achieved using the marginal sensitivity model (MSM) (Tan, 2006), a framework from causal sensitivity analysis that bounds the effect of unobserved confounding. However, the existing method for confounding-robust policy learning under the MSM (see (Kallus & Zhou, 2018a) for the conference paper and (Kallus & Zhou, 2021) for the journal version) has notable shortcomings. First, it must numerically optimize the worst-case effect on the regret function due to unobserved confounding. Such minimax optimization is based on inverse propensity weighted outcomes and hence unstable. Second, this method is statistically suboptimal: it lacks the property of semi-parametric efficiency and thus suffers from suboptimal variance properties.

In this paper, we address the above shortcomings by developing a novel *semi-parametrically efficient and sharp estimator for personalized off-policy learning under unobserved confounding*. Here,

semi-parametric efficiency means the unbiased estimator with the lowest possible variance. Our key novelties are the following: (i) We derive a *closed-form expression* for a *sharp bound on the value function* of a candidate policy under unobserved confounding.[1] As a result, we can thereby directly minimize our closed-form bound and, unlike existing works, avoid an unstable minimax optimization based on inverse propensity weighted outcomes. (ii) We propose an estimator that is semi-parametrically efficient. Hence, our estimator is the first to achieve the *lowest variance* among all unbiased estimators for our task.

Methodologically, we proceed as follows. We first derive a *sharp bound* on the value function for scenarios with unobserved confounding and, hence, avoid the unstable minimax optimization as in other methods. We then propose a novel *one-step bias-corrected estimator* to achieve semi-parametric efficiency and thus guarantee that our estimator has the lowest variance among all unbiased estimators. For this, we derive the corresponding efficient influence function of the sharp bound on the value function. We finally provide theoretical guarantees that minimizing our estimated sharp bound on the value function ensures that our method yields the *optimal* confounding-robust policy. Such guarantees are particularly crucial in high-stakes applications such as medicine or public policy, where unreliable policies can lead to harmful consequences.

Our work makes the following **contributions**[2]: (i) We propose a novel *efficient estimator for our sharp bound on the value function*. (ii) We derive an estimator for our bounds that is *semi-parametrically efficient*. (iii) We generalize our theoretical findings to the related task of *confounding-robust policy improvement*. (iv) Through extensive experiments using synthetic and real-world datasets, we show that our method consistently *outperforms* simple plug-in estimators and existing baselines.

## 2 RELATED WORK

We provide an overview of three literature streams particularly relevant to our work, namely, standard off-policy learning (i) with and (ii) without unobserved confounding as well as (iii) causal sensitivity analysis. We provide an extended related work in Appendix A (where we distinguish our work from other streams such as, e.g., unobserved confounding in reinforcement learning).

|  | Robust under unobserved conf. | Sharp bounds | Discrete treatments | Efficient for policy learning |
|---|---|---|---|---|
| Oprescu et al. (2023) | ✓ | ✓ | ✗ | ✗ |
| Swaminathan & Joachims (2015) | ✗ | ✗ | ✓ | ✗ |
| Athey & Wager (2021); Dudik et al. (2011) | ✗ | ✗ | ✓ | ✓ |
| Kallus & Zhou (2018a; 2021) | ✓ | ✗ | ✓ | ✗ |
| Efficient & sharp (ours) | ✓ | ✓ | ✓ | ✓ |

Table 1: Overview of related methods. Oprescu et al. (2023) is designed for a different task, and standard policy learners ignore the issue of unobserved confounding. Only Kallus & Zhou (2018a; 2021) deals with our setting, but provides neither sharp bounds nor an efficient estimator. Only our work can deal with unobserved confounding, discrete treatments, provides sharp bounds, and an efficient estimator.

**Off-policy learning under unconfoundedness:** Off-policy learning aims to optimize the policy value, which needs to be estimated from data. For this, there are three major approaches: (i) the direct method (DM) (Qian & Murphy, 2011) leverages estimates of the response functions; (ii) inverse propensity weighting (IPW) (Swaminathan & Joachims, 2015) re-weights the data such that in order to resemble samples under the evaluation policy; and (iii) the doubly robust method (DR) (Athey & Wager, 2021; Dudik et al., 2011). The latter is based on the efficient influence function of the policy value (Robins et al., 1994) and is efficient (Chernozhukov et al., 2018; van der Vaart, 1998).

Several works aim at improving the finite sample performance of these methods, for instance, via re-weighting (Kallus, 2018; 2021) or TMLE (Bibaut et al., 2019). Further, several methods have been proposed for off-policy learning in specific settings involving, for example, distributional robustness (Kallus et al., 2022), fairness (Frauen et al., 2024b), interpretability (Tschernutter et al., 2022), and continuous treatments (Kallus & Zhou, 2018b; Schweisthal et al., 2023). However, all of the works assume unconfoundedness and, therefore, do **not** account for unobserved confounding.

**Off-policy learning under unobserved confounding:** In scenarios with unobserved confounding, standard approaches for off-policy learning are biased (Kallus & Zhou, 2018a; 2021), which can lead to harmful decisions. The reason is that, under unobserved confounding, the policy value *cannot* be

---

[1] We use the term "sharp" as in earlier work from causal sensitivity analysis (Frauen et al., 2023): a valid upper (lower) bound of a causal quantity is *sharp* if there does not exist another valid bound that is strictly smaller (larger).

[2] Code is available at `https://github.com/konstantinhess/Efficient_sharp_policy_learning`.

identified from observational data. As a remedy, previous works leverage causal sensitivity analysis or related methods to obtain bounds on the unidentified policy value (Bellot & Chiappa, 2024; Guerdan et al., 2024; Huang & Wu, 2024; Joshi et al., 2024; Namkoong et al., 2020; Zhang & Bareinboim, 2024), which can then be used to learn an optimal worst-case policy. Optimizing such bounds is often termed "confounding-robust policy learning". However, these works do *not* consider sharp bounds under unobserved confounding and do *not* provide semi-parametrically efficient estimators.

Closest to our work is (Kallus & Zhou, 2018a) with an extended version published in (Kallus & Zhou, 2021). Therein, the authors propose a method for confounding-robust policy improvement, yet with two notable shortcomings: (i) it is *not* based on closed-form solutions for the bounds, and (ii) it is *not* based on a semi-parametrically efficient estimator of these bounds. Therefore, Kallus & Zhou (2018a; 2021) require solving a minimax optimization problem that is relies in inverse propensity weighted outcomes, which is *unstable*. Further, their estimator is suboptimal because it fails to achieve semi-parametric efficiency, meaning it does *not* achieve the lowest-possible variance among all unbiased estimators. We provide a thorough discussion on the work by Kallus & Zhou (2018a; 2021) and a detailed comparison to our method in Supplement B.

**Causal sensitivity analysis:** Causal sensitivity analysis (Cornfield et al., 1959) allows practitioners to account for unobserved confounding by using so-called sensitivity models (Jin et al., 2022; Rosenbaum, 1987), which incorporate domain knowledge on the strength of unobserved confounding. As a result, the sensitivity model allows to obtain bounds on a causal quantity of interest, which, if the sensitivity model is correctly specified, can then be used for consequential decision-making (Jesson et al., 2021).

A prominent sensitivity model is the MSM (Tan, 2006). The MSM gained popularity in recent years and, for instance, was used to obtain bounds on the conditional average treatment effect (CATE) through machine learning (Jesson et al., 2021; Kallus et al., 2019; Yin et al., 2022). Works like Aronow & Lee (2013); Miratrix & Wager (2018); Zhao et al. (2019) target mean effects under the MSM, rely on Hájek-style estimators which have finite-sample bias, and are not designed for policy learning. Yadlowsky et al. (2022) study ATE bounds, but their framework is limited to *binary* treatments. Finally, Bruns-Smith & Zhou consider *dynamic* policies in Markov decision processes and must estimate ratios of transition probabilities, which yields a fundamentally different estimator that is not sharp for action spaces with more than two actions. Only recently, sharp bounds on the CATE have been derived (Bonvini et al., 2022; Dorn & Guo, 2022; Frauen et al., 2024a; 2023; Jin et al., 2023). Other works have considered the estimation of such bounds (Dorn et al., 2024; Oprescu et al., 2023), or instruments for partial identification Schweisthal et al. (2025). However, these works only consider causal sensitivity analysis for CATE but **not** policy learning. Further, their works are limited to **binary** treatments. In contrast, our method can handle *discrete* treatments, and, therefore, requires entirely different influence functions.

**Research gap:** To the best of our knowledge, we are the first to derive a *semi-parametrically efficient estimator for a sharp bound on the value function using the MSM*. Thereby, we enable optimal confounding-robust off-policy learning.

## 3 PROBLEM SETUP

**Data:** Let $Y \in \mathcal{Y} \subset \mathbb{R}$ be our outcome of interest, such as the health condition of a patient. We follow the convention that w.l.o.g. *lower* values correspond to *better* outcomes (Kallus & Zhou, 2018a). Further, let $X \in \mathcal{X} \subset \mathbb{R}^{d_x}$ denote covariates that contain additional information, such as age, gender, or disease-related information. Finally, let $A \in \mathcal{A} = \{0, 1, \dots, d_a - 1\}$ be the assigned treatment (or action). Note that we do not restrict our setting to binary treatments but allow for arbitrary, discrete treatments. For the product space, we use $\mathcal{D} = \mathcal{Y} \times \mathcal{X} \times \mathcal{A}$. In the following, we assume that we have access to an observational dataset $\mathcal{D}_n = \{(Y_i, X_i, A_i)\}_{i=1}^n$ that consists of $n$ i.i.d. copies of $(Y, X, A) \in \mathcal{D}$.

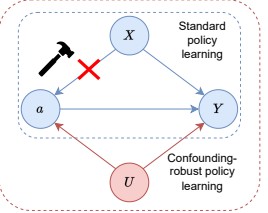

Figure 1: We can only block backdoor paths for observed confounders $X$. Hence, under unobserved confounding $U$, we cannot point-identify $V(\pi)$.

**Policy value:** Policy learning aims to find the best policy for assigning treatments, given covariates. Formally, a *policy* $\pi(a \mid x)$ is a conditional probability mass function $\pi : \mathcal{A} \times \mathcal{X} \to [0, 1]$ with $\sum_{a \in \mathcal{A}} \pi(a \mid x) = 1$, corresponding to the probability of receiving treatment $A = a$, given covariates

$X = x$. The *value $V(\pi)$* of a policy is defined as $V(\pi) = \mathbb{E}\Big[\sum_{a \in \mathcal{A}} \pi(a \mid X) Y[a]\Big]$, where $Y[a]$ denotes the potential outcome for $Y$ when intervening on treatment $A = a$ (Neyman, 1923; Rubin, 1978). Hence, the policy value $V(\pi)$ is the expected average potential outcome when adhering to the policy $\pi$.

**Standard off-policy learning:** Off-policy learning aims to find a policy $\pi$ that has the best policy value among all $\pi \in \Pi$ for some policy class $\Pi$. Of note, it is standard in the literature (Frauen et al., 2024b; Hatt et al., 2022; Kallus & Zhou, 2018a) to restrict the analysis to policy classes $\Pi$ with finite complexity such as neural networks.

The value function is identifiable under the following three assumptions (Rubin, 1978): (i) *Consistency:* $Y[A] = Y$; (ii) *Strong overlap:* There exists $c \in (0, 1)$ such that $c \leq p(A = a \mid X = x) \leq 1 - c \; \forall \, a \in \mathcal{A}, x \in \mathcal{X}$; (iii) *Unconfoundedness:* $Y[a] \perp\!\!\!\perp A \mid X \; \forall \, a \in \mathcal{A}$. Then, the policy value is identified from the observational data via $V(\pi) = \mathbb{E}\Big[\sum_{a \in \mathcal{A}} \pi(a \mid X) Q(a, X)\Big]$, where $Q(a, x) = \mathbb{E}[Y \mid X = x, A = a]$ is the conditional average potential outcome function.

The optimal policy can then be learned via

$$\pi^*_{\text{standard}} = \arg\min_{\pi \in \Pi} \hat{V}(\pi), \tag{1}$$

where $\hat{V}(\pi)$ is an estimator of the identified policy value. Recall that we follow the convention in (Kallus & Zhou, 2018a; 2021) that lower $Y$ are better, so we aim to *minimize* the value function.

**Allowing for unobserved confounding:** The assumption of (iii) *unconfoundedness* is problematic and often unrealistic (Hemkens et al., 2018): Unconfoundedness requires that the observed covariates $X$ capture *all* factors that affect both treatment choice and outcome. In this work, we do **not** rely on the *unconfoundedness* assumption, which is restrictive and oftentimes unrealistic. Instead, we allow for *unobserved confounding*, which we denote by a random variable $U \in \mathcal{U} \subset \mathbb{R}$ (see Figure 2).

Importantly, under unobserved confounding, we cannot point-identify the value function $V(\pi)$. Instead, we aim to *partially* identify the value function $V(\pi)$ by leveraging causal sensitivity analysis. Specifically, we adopt the MSM (Tan, 2006) to bound the ratio between the *nominal propensity score*

$$e(a, x) = p(A = a \mid X = x), \tag{2}$$

which can be estimated from data $\mathcal{D}_n$ and the *true propensity score*

$$e(a, x, u) = p(A = a \mid X = x, U = u), \tag{3}$$

which is fundamentally unobserved. Formally, the MSM assumes

$$\Gamma^{-1} \leq \frac{e(a, x)}{1 - e(a, x)} \frac{1 - e(a, x, u)}{e(a, x, u)} \leq \Gamma \tag{4}$$

for some $\Gamma \geq 1$ that can be chosen by domain domain knowledge (Frauen et al., 2023; Kallus et al., 2019) or data-driven heuristics (Hatt et al., 2022) (see Supplement E).

Intuitively, $\Gamma$ close to $1$ implies that the impact of unobserved variables $U$ on the treatment decision is small, whereas a large $\Gamma$ means that observed variables $X$ do not contain sufficient information to fully capture the treatment decision. In particular, $\Gamma = 1$ implies that the true propensity score coincides with the nominal propensity score. Hence, there is no unobserved confounding and our scenario simplifies to the naïve unconfoundedness setting. Conversely, if we let $\Gamma > 1$, the true and the nominal propensity scores differ, and, therefore, we account for additional unobserved confounding.

Formally, the marginal sensitivity model gives rise to a set of distributions $\mathcal{P}(\Gamma)$ over $\mathcal{D} \times \mathcal{U}$ that are compatible with the constraints in Equation 4. This set is defined as

$$\mathcal{P}(\Gamma) = \left\{ \tilde{p} \in \mathcal{P}(\mathcal{D} \times \mathcal{U}) : \int_{\mathcal{U}} \tilde{p}(d, u) \, \mathrm{d}u = p(d) \forall d \in \mathcal{D}, \; \Gamma^{-1} \leq \frac{\tilde{e}(A, X)}{1 - \tilde{e}(A, X)} \frac{1 - \tilde{e}(A, X, U)}{\tilde{e}(A, X, U)} \leq \Gamma \right\}, \tag{5}$$

where $\mathcal{P}(\mathcal{D} \times \mathcal{U})$ is the set of all possible joint distributions of the observables and the unobserved confounders, and where the nominal propensity score $\tilde{e}(A, X)$ and the true propensity score $\tilde{e}(A, X, U)$ result from $\tilde{p}$ as in Equation 2 and Equation 3, respectively.

Different from standard off-policy learning under the unconfoundedness assumption, we can *not* point-identify the value function, and, hence, optimizing the objective in Equation 1 is *biased*. Instead, we need to account for the worst-case scenario that can occur under unobserved confounding. That is, we are interested in: *Which policy yields the optimal value under the worst-case unobserved confounding, given our sensitivity constraints?*

**Objective:** Formally, the optimal confounding-robust policy $\pi^*$ is the solution to the minimax problem

$$\pi^* = \arg\min_{\pi \in \Pi} \sup_{\tilde{p} \in \mathcal{P}(\Gamma)} V(\pi). \tag{6}$$

However, the existing method (Kallus & Zhou, 2018a; 2021) for our task has key limitations: (i) It requires directly solving the minimax optimization problem, which can be unstable due to inverse propensity weighting. Instead, we later derive a *closed-form expression for the inner supremum* (i.e., an upper bound), which reduces Equation 6 to a simple minimization task. (ii) This method is *not* semi-parametrically efficient, thus leading to suboptimal finite-sample performance. As a remedy, we later derive an estimator that is *semi-parametrically efficient*.

## 4 SHARP BOUNDS AND EFFICIENT ESTIMATION

In this section, we introduce our estimator for sharp bounds of the value function under unobserved confounding. For this, we first derive a *closed-form solution* for the sharp bounds of the value function (Section 4.1), which directly solves the inner maximization in Equation 6. Then, we present our estimator for these bounds (Section 4.2), which is based on non-trivial derivations of the efficient influence function to offer *semi-parametric efficiency*. Further, we provide *learning guarantees* when optimizing the bounds of the value function (Section 4.3). Finally, we propose an *extension of our method* for scenarios where the aim is to optimize the relative improvement of a policy over a given baseline policy such as the standard of care in medicine (Section 4.4).

### 4.1 SHARP BOUNDS FOR THE VALUE FUNCTION

We now derive our sharp bound for the value function under unobserved confounding, given our sensitivity constraints $\mathcal{P}(\Gamma)$ in Equation 5. Recall that our aim is to *minimize* the value function $V(\pi)$, and, hence, we are interested in an *upper bound* for $V(\pi)$. That is, we seek to find the value function in the worst-case confounding scenario under the MSM, which is given by $V^{+,*}(\pi) = \sup_{\tilde{p} \in \mathcal{P}(\Gamma)} V(\pi)$.

By definition, a closed-form solution to this maximization problem ensures that (i) the bound is *valid*, i.e., $V^{+,*}(\pi) \geq V(\pi)$ for all $\tilde{p} \in \mathcal{P}(\Gamma)$, and that (ii) the bound is *sharp*, i.e., there does **not** exist a valid upper bound $V^{+,\dagger}(\pi)$ such that $V^{+,\dagger}(\pi) < V^{+,*}(\pi)$.

In order to derive $V^{+,*}(\pi)$, we first introduce the *conditional average potential outcome* function

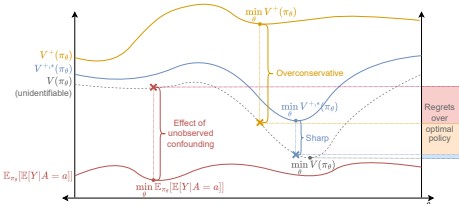

Figure 2: Under unobserved confounding, the value function is unidentifiable, and the ground-truth optimal policy is unknown. Ignoring unobserved confounding can lead to a policy with *large regret*, and may introduce harm. Further, optimizing w.r.t. a suboptimal bound can lead to an *overconservative* policy. Instead, we seek to find the optimal confounding-robust policy by minimizing a *sharp bound* on the worst-case effect of unobserved confounding.

$$Q(a, x) = \mathbb{E}[Y[a] \mid X = x], \tag{7}$$

which is the expected potential outcome for treatment $A = a$, given covariate information $X = x$. Importantly, because we do **not** make Assumption (iii) of *unconfoundedness*, the quantity $Q(a, x)$ is not point-identified.

We now state our first theorem, which provides a sharp upper bound of the value function under our sensitivity constraints $\mathcal{P}(\Gamma)$. Further, we also provide the sharp lower bound $V^{-,*} = \inf_{\tilde{p} \in \mathcal{P}(\Gamma)} V(\pi)$, which we later need for our extensions in Section 4.4.

**Proposition 4.1.** *Let $Q^{+,*}(a,x) = \sup_{\tilde{p} \in \mathcal{P}(\Gamma)} Q(a,x)$ and $Q^{-,*}(a,x) = \inf_{\tilde{p} \in \mathcal{P}(\Gamma)} Q(a,x)$ be the sharp upper and lower bound for the conditional average potential outcome, respectively, given our sensitivity constraints $\mathcal{P}(\Gamma)$. Then, the sharp upper bound $\sup_{\tilde{p} \in \mathcal{P}(\Gamma)} V(\pi) = V^{+,*}(\pi)$ and the sharp lower bound $\inf_{\tilde{p} \in \mathcal{P}(\Gamma)} V(\pi) = V^{-,*}(\pi)$ for the value function $V(\pi)$ are given by*

$$V^{\pm,*}(\pi) = \int_{\mathcal{X}} \sum_a Q^{\pm,*}(a,x)\pi(a \mid x)\,\mathrm{d}p(x). \tag{8}$$

*Proof.* See Supplement F.1. □

Our above derivation of the closed-form solution has a crucial advantage over existing works (Kallus & Zhou, 2018a; 2021): we avoid an unstable minimax optimization that is based on inverse propensity weighted outcomes, and, instead, we can directly work with $V^{+,*}(\pi)$, which simplifies Equation 6 to $\pi^* = \arg\min_{\pi \in \Pi} V^{+,*}(\pi)$. As a result, we have reduced the original minimax problem to a much simpler *minimization task*.

## 4.2 Semi-parametrically efficient estimator for the sharp upper bound

In this section, we derive a semi-parametrically efficient estimator for our sharp upper bound $V^{+,*}(\pi)$ of the value function $V(\pi)$. semi-parametrically efficient estimators are desirable because they achieve the *lowest possible variance among all unbiased estimators* (Hines et al., 2022; Kennedy, 2022).

In order to derive such an estimator of $V^{+,*}(\pi)$, we first need to decompose the estimand $Q^{\pm,*}(a,x)$ in Proposition 4.1.

**Definition 4.2** ((Dorn & Guo, 2022; Frauen et al., 2023))**.** Sharp bounds $Q^{\pm,*}(a,x)$ of the conditional average potential outcome $Q(a,x)$ function are given by

$$Q^{\pm,*}(a,x) = c^{\mp}(a,x)\underline{\mu}^{\pm}(a,x) + c^{\pm}(a,x)\bar{\mu}^{\pm}(a,x), \tag{9}$$

where we let

$$c^{\pm}(a,x) = b^{\pm}e(a,x) + \Gamma^{\pm 1}, \; b^{\pm} = (1 - \Gamma^{\pm 1}) \tag{10}$$

and

$$\underline{\mu}^{\pm}(a,x) = \mathbb{E}[Y\underline{\Delta}^{\pm}(Y,A,X) \mid X = x, A = a], \quad \bar{\mu}^{\pm}(a,x) = \mathbb{E}[Y\bar{\Delta}^{\pm}(Y,A,X) \mid X = x, A = a] \tag{11}$$

with

$$\underline{\Delta}^{\pm}(y,a,x) = \mathbb{1}_{\{y \le F_{x,a}^{-1}(\alpha^{\pm})\}}, \qquad\qquad \bar{\Delta}^{\pm}(y,a,x) = \mathbb{1}_{\{y \ge F_{x,a}^{-1}(\alpha^{\pm})\}}, \tag{12}$$

where $\alpha^+ = \Gamma/(1+\Gamma)$ and $\alpha^- = 1/(1+\Gamma)$, and where $F_{x,a}^{-1}(q)$ is the conditional quantile function

$$F_{x,a}^{-1}(q) = \inf\{y \in \mathcal{Y} : \; p(Y \le y \mid X = x, A = a) \ge q\}. \tag{13}$$

In order to achieve semi-parametric efficiency for the sharp upper bound $V^{+,*}(\pi)$, we need to carefully take into account the nuisance functions in Equation 8 and Equation 9, respectively. That is, the key difficulty lies in that $V^{+,*}(\pi)$ depends on several nuisance functions

$$\eta = \{e(a,x), F_{a,x}^{-1}(\alpha^{\pm}), \bar{\mu}^{\pm}(a,x), \underline{\mu}^{\pm}(a,x)\}. \tag{14}$$

If we followed a naïve plug-in approach (i.e., if we estimated $\hat{\eta}$ from data $\mathcal{D}_n$ and plugged them into Equation 8 and thus Equation 9), our final estimator $\hat{V}^{+,*}(\pi)$ would suffer from **first-order bias** due to estimation errors in the nuisance functions. As a remedy, we present a *one-step bias-corrected* estimator. That is, we estimate the first-order bias and subtract it from our plug-in estimate (Hess et al., 2026; Frauen et al., 2025a; Kennedy, 2022; van der Vaart, 1998).

**Theorem 4.3.** *Let $\mathbb{E}[|Y|] < \infty$ and $p(y \mid x, a)$ admit a continuous density bounded away from zero in a neighborhood of $F_{x,a}^{-1}(\alpha^+)$. This holds, for example, if the density $p(y \mid x, a)$ is strictly positive and continuous. Then, an estimator for the sharp upper bound of the value function is given by*

$$
\begin{aligned}
\hat{V}^{+,*}(\pi) = \mathbb{P}_n \Big\{ &\sum_a \pi_{a,X} \Big[ \hat{Q}_{a,X}^{+,*} - \hat{e}_{a,X} \Big( b^- \underline{\hat{\mu}}_{a,X}^+ + b^+ \hat{\bar{\mu}}_{a,X}^+ \Big) \Big] + \pi_{A,X} \Big( b^- \underline{\hat{\mu}}_{A,X}^+ + b^+ \hat{\bar{\mu}}_{A,X}^+ \Big) \\
&+ \frac{\pi_{A,X}}{\hat{e}_{A,X}} \Big[ \Big( \hat{c}_{A,X}^- - \hat{c}_{A,X}^+ \Big) \Big( \hat{F}_{X,A}^{-1}(\alpha^+)(\underline{\hat{\Delta}}_{Y,A,X}^+ - \alpha^+) \Big) \\
&+ \hat{c}_{A,X}^- \Big( Y \underline{\hat{\Delta}}_{Y,A,X}^+ - \underline{\hat{\mu}}_{A,X}^+ \Big) + \hat{c}_{A,X}^+ \Big( Y \hat{\bar{\Delta}}_{Y,A,X}^+ - \hat{\bar{\mu}}_{A,X}^+ \Big) \Big] \Big\}.
\end{aligned} \tag{15}
$$

*If, on top, $\mathbb{E}[|Y|^2] < \infty$, the above estimator is **semi-parametrically efficient**.*

*Proof.* See Supplement F.2. □

We now have a *semi-parametrically efficient estimator* for the sharp upper bound of the value function under unobserved confounding. Algorithm 1 presents a flexible procedure to learn confounding-robust policies for parametric policy classes $\Pi_\theta$ (e.g., neural networks). For simplicity, Algorithm 1 only uses a single spit, but we emphasize that cross-fitting can be used to increase sample efficiency.

### 4.3 LEARNING GUARANTEES

In this section, we provide asymptotic learning guarantees in the form of generalization bounds when learning the confounding-robust policy $\pi$ via our Algorithm 1. Of note, it is *not* obvious that minimizing the *estimated* sharp upper bound $\hat{V}^{+,*}(\pi)$ provides a meaningful, confounding-robust policy $\pi^*$. Hence, we provide learning guarantees where we show that, with high probability, *minimizing our estimated sharp upper bound yields the optimal policy*.

For this, we show that minimizing the *estimated* sharp upper bound $\hat{V}^{+,*}(\pi)$ with respect to $\pi$ indeed minimizes the true, unknown value function $V(\pi)$ on population level. Fortunately, our method only requires one additional assumption, namely, boundedness of the outcome $|Y| \leq C_y$. This is a very mild restriction and reasonable in practice.

We express the flexibility of our policy class $\Pi$ in terms of the Rademacher complexity $\mathcal{R}_n(\pi)$, which is a common choice in the literature (Athey & Wager, 2021; Frauen et al., 2024b; Hatt et al., 2022; Kallus & Zhou, 2018a). For example, the class of neural networks with bounded

---

**Algorithm 1** Confounding-robust policy learning.

**Input:** Data $\mathcal{D}_n = \{(Y_i, A_i, X_i)\}_{i=1}^n$, sensitivity parameter $\Gamma \geq 1$, sample split $\rho \in (0,1)$, learning rate $\lambda$, parametric policy class $\Pi_\theta$, training iterations $K$

**Output:** $\hat{V}^{+,*}(\pi)$

1: Perform sample split $\mathcal{D}_{\lceil \rho n \rceil}^\eta, \mathcal{D}_{\lfloor (1-\rho)n \rfloor}^{V^{+,*}}$ (Employ cross-fitting to improve sample-efficiency.)
2: Estimate nuisance functions $\hat{\eta}$ on $\mathcal{D}_{\lceil \rho n \rceil}^\eta$
3: Evaluate $\hat{\eta}$ on $\mathcal{D}_{\lfloor (1-\rho)n \rfloor}^{V^{+,*}}$
4: Initialize policy $\pi_\theta^{(0)} \in \Pi_\theta$
5: **for** $k = 0$ to $K-1$ **do**
6:     Estimate $V^{+,*}(\pi_\theta^{(k)})$ as in Equation 38 (using evaluated $\hat{\eta}$)
7:     Update policy parameters:
8:       $\theta^{(k+1)} \leftarrow \theta^{(k)} - \lambda \nabla_\theta V^{+,*}(\pi_\theta^{(k)})$
9: **end for**
10: **Return:** Robust policy $\pi^* \leftarrow \pi_\theta^{(K)}$

---

size and smoothness, e.g., with fixed depth, bounded width, and weights controlled by simple norm constraints (so the networks are Lipschitz and outputs bounded), has Rademacher complexity $\mathcal{R}_n(\Pi) \in \mathcal{O}(n^{-1/2})$.

**Theorem 4.4.** *Assume $|Y| \leq C_y$. Let $C_v := 2C_y(1 + \Gamma^{-1} + \Gamma)$ and let $R_n(\Pi)$ denote the empirical Rademacher complexity of the policy class $\Pi$. Then, with probability at least $1 - \delta$,*

$$\sup_{\pi \in \Pi} \left\{ V(\pi) - \hat{V}^{+,*}(\pi) \right\} \leq 2C_v \left( R_n(\Pi) + \frac{5}{2}\sqrt{\frac{1}{2n}\log\frac{2}{\delta}} \right). \tag{16}$$

*Equivalently, on the same high-probability event,* simultaneously for all $\pi \in \Pi$,

$$V(\pi) \leq \hat{V}^{+,*}(\pi) + 2C_v \left( R_n(\Pi) + \frac{5}{2}\sqrt{\frac{1}{2n}\log\frac{2}{\delta}} \right). \tag{17}$$

*Proof.* See Supplement F.3. □

The above Theorem 4.4 has the following implication: given our sensitivity constraints $\mathcal{P}(\Gamma)$, our estimated sharp upper bound $\hat{V}^{+,*}(\pi)$ correctly bounds the true, unknown value function $V(\pi)$ on population level with high probability. Therefore, given sufficient data $\mathcal{D}_n$, minimizing $\hat{V}^{+,*}(\pi)$ with respect to $\pi$ also minimizes $V(\pi)$ and, hence, yields the optimal $\pi^*$.

In sum, we have derived (i) a novel *sharp upper bound of the value function*, which circumvents unstable minimax optimization based on inverse propensity weighted outcomes. Further, we have proposed (ii) an estimator for this bound that is *semi-parametrically efficient*, i.e., an unbiased estimator with the lowest possible variance. Finally, we have derived (iii) *learning guarantees*, which show that minimizing our estimated bound via Algorithm 1 indeed optimizes the true, unknown population value.

## 4.4 EXTENSION TO POLICY IMPROVEMENT

Our main results from above focus on optimizing the value function $V(\pi)$, which is common in practice (e.g., Dudik et al., 2011; Hatt et al., 2022). However, in some scenarios, an established baseline policy $\pi_0$ may be available; then, one may aim to make a small relative improvement yet with certain guarantees. This setting is commonly termed as *policy improvement* (Kallus & Zhou, 2018a; 2021; Laroche et al., 2019; Thomas et al., 2015).

Hence, we no longer aim to minimize bounds on the value function $V(\pi)$ but, instead, bounds on the regret of a candidate policy against a baseline policy (Kallus & Zhou, 2018a). Specifically, we can define the *regret* of policy $\pi$ over baseline $\pi_0$ as $R_{\pi_0}(\pi) = V(\pi) - V(\pi_0)$. Hence, a negative regret implies that policy $\pi$ improves upon $\pi_0$. Importantly, the optimal confounding-robust policy $\pi^*$ in Equation 6 can also be defined as the policy $\pi$ that achieves the best relative improvement over baseline $\pi_0$ in the worst-case scenario, that is,

$$\pi^* = \arg\min_{\pi \in \Pi} \ \sup_{\tilde{p} \in \mathcal{P}(\Gamma)} \ R_{\pi_0}(\pi). \tag{18}$$

This definition is *equivalent* to Equation 6. Nevertheless, the above objective may be preferred in practice when aiming at policy improvement.

We now show in the following three corollaries that our results directly generalize to policy improvement. First, we provide a closed-form solution for an upper bound of the regret function $R_{\pi_0}(\pi)$, given our sensitivity constraints $\mathcal{P}(\Gamma)$.

**Corollary 4.5.** *An upper bound for the regret function $R_{\pi_0}(\pi)$ is given by $R_\pi^+(\pi) = \int_{\mathcal{X}} \sum_a \left( Q^{+,*}(a,x)\pi(a \mid x) - Q^{-,*}(a,x)\pi_0(a \mid x) \right) \mathrm{d}p(x)$.*

*Proof.* See Supplement F.4. □

Next, we derive a semi-parametrically efficient, one-step bias-corrected estimator, which is based on the efficient influence function.

**Corollary 4.6.** *A semi-parametrically efficient estimator for the upper regret bound is given by*

$$\hat{R}_{\pi_0}^+(\pi) = \sum_{\pm \in \{-,+\}} \pm \mathbb{P}_n \Big\{ \sum_a \pi_{a,X}^\pm \Big[ \hat{Q}_{a,X}^{\pm,*} - \hat{e}_{a,X} \Big( b^\mp \hat{\underline{\mu}}_{a,X}^\pm + b^\pm \hat{\bar{\mu}}_{a,X}^\pm \Big) \Big]$$

$$+ \pi_{A,X}^\pm \Big( b^\mp \hat{\underline{\mu}}_{A,X}^\pm + b^\pm \hat{\bar{\mu}}_{A,X}^\pm \Big) + \frac{\pi_{A,X}^\pm}{\hat{e}_{A,X}} \Big[ \Big( \hat{c}_{A,X}^\mp - \hat{c}_{A,X}^\pm \Big) \Big( \hat{F}_{X,A}^{-1}(\alpha^\pm)(\hat{\underline{\Delta}}_{Y,A,X}^\pm - \alpha^\pm) \Big)$$

$$+ \hat{c}_{A,X}^\mp \Big( Y \hat{\underline{\Delta}}_{Y,A,X}^\pm - \hat{\underline{\mu}}_{A,X}^\pm \Big) + \hat{c}_{A,X}^\pm \Big( Y \hat{\bar{\Delta}}_{Y,A,X}^\pm - \hat{\bar{\mu}}_{A,X}^\pm \Big) \Big] \Big\},$$

*where we let $\pi^+ = \pi$ and $\pi^- = \pi_0$ for readability.*

*Proof.* See Supplement F.5. □

Finally, we provide improvement guarantees: given a baseline policy $\pi_0$ (e.g., the standard of care), if the empirical estimator $\hat{R}_{\pi_0}(\pi)^+$ is *negative*, which we can check by evaluating it, we are *guaranteed* that $\pi$ improves upon $\pi_0$ and introduces *no harm*.

**Corollary 4.7.** *Under the same assumption as in Theorem 4.4, for any policy $\pi \in \Pi$ and baseline policy $\pi_0 \in \Pi$, it holds, with probability $1 - \delta$, that $R_{\pi_0}(\pi) \leq \hat{R}_{\pi_0}^+(\pi) + 4C_v \Big( \mathcal{R}_n(\Pi) + \frac{5}{2} \sqrt{\frac{1}{2n} \log \left( \frac{2}{\delta} \right)} \Big).$*

*Proof.* See Supplement F.6. □

| | $\Gamma^* = 2$ | $\Gamma^* = 4$ | $\Gamma^* = 6$ | $\Gamma^* = 8$ | $\Gamma^* = 10$ | $\Gamma^* = 12$ | $\Gamma^* = 14$ | $\Gamma^* = 16$ |
|---|---|---|---|---|---|---|---|---|
| Standard IPW estimator | $-\mathbf{1.31} \pm \mathbf{0.02}$ | $-0.60 \pm 0.15$ | $-0.09 \pm 0.01$ | $-0.07 \pm 0.01$ | $-0.06 \pm 0.01$ | $-0.06 \pm 0.01$ | $-0.05 \pm 0.01$ | $-0.03 \pm 0.01$ |
| Standard DR estimator | $-1.30 \pm 0.04$ | $-0.71 \pm 0.02$ | $-0.18 \pm 0.13$ | $-0.07 \pm 0.01$ | $-0.07 \pm 0.01$ | $-0.06 \pm 0.01$ | $-0.05 \pm 0.01$ | $-0.04 \pm 0.01$ |
| Kallus & Zhou (2018; 2021) | $-1.21 \pm 0.10$ | $-0.70 \pm 0.06$ | $-0.40 \pm 0.06$ | $-0.22 \pm 0.04$ | $-0.16 \pm 0.02$ | $-0.14 \pm 0.02$ | $-0.10 \pm 0.01$ | $-0.08 \pm 0.01$ |
| **Efficient + sharp estimator** (ours) | $-1.12 \pm 0.08$ | $-\mathbf{1.00} \pm \mathbf{0.08}$ | $-\mathbf{0.89} \pm \mathbf{0.13}$ | $-\mathbf{0.66} \pm \mathbf{0.14}$ | $-\mathbf{0.64} \pm \mathbf{0.14}$ | $-\mathbf{0.58} \pm \mathbf{0.17}$ | $-\mathbf{0.50} \pm \mathbf{0.20}$ | $-\mathbf{0.30} \pm \mathbf{0.22}$ |
| Absolute improvement | $+0.19$ | $-0.29$ | $-0.49$ | $-0.44$ | $-0.48$ | $-0.44$ | $-0.40$ | $-0.22$ |

Table 2: **Varying confounding strength.** We vary the confounding parameter $\Gamma^*$ in the DGP along with the sensitivity parameter $\Gamma$ in both our efficient estimator and the baseline (Kallus & Zhou, 2018a; 2021). Then, we report the regret over a randomized policy (*lower values are better*). As confounding increases, our estimator is the only method that is robust and thus performs best.

## 5 EXPERIMENTS

In the following, we evaluate the performance of our method against: (1) the minimax optimization approach by Kallus & Zhou (2018a; 2021) and standard methods for policy learning, namely, (2) the IPW estimator (Swaminathan & Joachims, 2015) and (3) the DR estimator (Athey & Wager, 2021; Dudik et al., 2011). Importantly, the approach by Kallus & Zhou (2018a; 2021) is the **only** baseline that can deal with confounding-robust policy learning with the MSM and thus the **only** baseline for our task. To ensure a *fair comparison*, we use the same neural instantiations for all models in terms of (i) the nuisance functions $\hat{\eta}$ and (ii) the policy $\pi_\theta$ (see Supp. G). All results are averaged over 10 seeds.

• **Synthetic Data:** As is standard in causal inference literature (Hess & Feuerriegel, 2025; Hess et al., 2024; Kallus et al., 2019), we evaluate our method on synthetic data in order to have access to ground-truth counterfactuals. Here, we use an established data-generating process from the literature (Kallus et al., 2019): First, we simulate observed confounders $X \sim \text{Unif}[-2, 2]$ and unobserved confounders $U \sim \text{Ber}(1/2)$. The potential outcomes $Y[a]$ are then given by $Y[a] = (2a-1)X + (2a-1) - 2\sin(2(2a-1)X) - 2(2U-1)(1+0.5X) + \varepsilon$, where $\varepsilon \sim \mathcal{N}(0,1)$ is random noise. Further, we assume a binary treatment, i.e., $d_a = 2$. For this, we first fix a ground-truth $\Gamma^*$. Then, we let the *true propensity score* be given by $e(1, x, u) = \frac{u}{\rho(x; 1/\Gamma^*)} + \frac{1-u}{\rho(x; \Gamma^*)}$, where $\rho(x; \gamma) = 1 + (1/e(1, x) - 1)\gamma$, and $e(1, x) = \sigma(0.75x + 0.5)$ is the nominal propensity score.

**Varying confounding strength:** First, we demonstrate the performance of our method for increasing levels of unobserved confounding. For this, we increase the confounding parameter $\Gamma^*$ in the data-generating process. We compare the regret of each method over a randomized baseline policy. In our method and (Kallus & Zhou, 2018a; 2021), we set the sensitivity parameter $\Gamma$ equal to $\Gamma^*$.

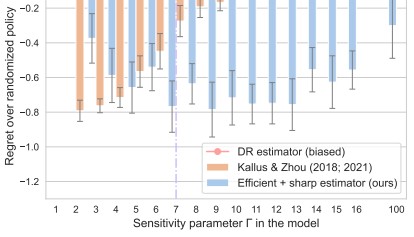

Figure 3: **Robustness analysis.** We set $\Gamma^* = 7$ in the data-generating process but use mis-specified sensitivity parameters $\Gamma$ in our estimator (i.e., $\Gamma = 7$ is correctly specified, while $\Gamma \neq 7$ is mis-specified). We report the regret over a randomized policy *(lower values are better)*. Our estimator significantly improves upon the standard DR estimator, even for a completely mis-specified $\Gamma$.

Our results are shown in **Table** 2: (i) As expected, the standard methods for off-policy learning (i.e., a standard IPW estimator and a standard DR estimator) perform well for zero to very low levels of confounding. However, the standard methods are *biased* and thus become ineffective for $\Gamma^* > 1$. (ii) The method by Kallus & Zhou (2018a; 2021) performs well under low levels of confounding. Yet, the performance quickly deteriorates. (iii) Our proposed method performs clearly best for increasing $\Gamma^*$. Here, our method achieves a *relative performance gain by up to a factor of* $4$.

**Robustness analysis:** Next, we show that our method is robust to mis-specification of the sensitivity parameter $\Gamma$. We thus fix the confounding strength to $\Gamma^* = 7$ in the DGP. We increase $\Gamma$ from $1$ (which corresponds to unconfoundedness) up to $100$ (which mirrors almost assumption-free bounds). We again compute the regret of our learned policy over a randomized baseline policy to showcase the improvement. **Figure** 3 shows that our approach yields *robust results even for mis-specified* $\Gamma$ (i.e., when $\Gamma \neq 7$). In contrast, the method by Kallus & Zhou (2018a; 2021) improves for mis-specified values $\Gamma < \Gamma^*$, but performance *quickly deteriorates* for larger $\Gamma$, which renders it *unreliable*. This behavior is expected, as their method only minimizes a bound on the regret over a baseline policy, and quickly reverts to the baseline when the specified $\Gamma$ increases (see Supplement B for a detailed

discussion). Further, even under the (almost) no assumptions constraint of $\Gamma = 100$, our method provides significant improvements over the biased DR estimator.

**Semi-parametrically efficient estimation:** Finally, we show the benefits of our efficient estimation strategy over simple plug-in estimators. A semi-parametrically efficient estimator is the unbiased estimator with the lowest variance (Kennedy, 2022; van der Vaart, 1998). Hence, policies based on efficiently estimated bounds are learned better in low sample settings than those based on plug-in approaches. Therefore, we report the performance when we vary the number of training samples $\mathcal{D}_n$. Here, we compare our method against a naïve plug-in estimator of our sharp bounds based on Proposition 4.1. We again report the regret over a randomized baseline policy. **Figure** 4 shows that our method performs better in low sample settings, and achieves larger performance gains when increasing the sample size.

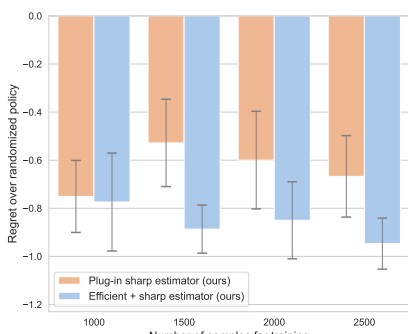

Figure 4: **Property of semi-parametrically efficient estimation.** We compare our efficient estimator with a simple plug-in estimator of our sharp upper bound from Proposition 4.1. We report the regret over a randomized policy *(lower values are better)*. Our efficient estimator leads to a lower regret and benefits from increasing sample size due to its optimal estimation properties.

• **Real-world medical data:** We evaluate our method on a real-world medical case study. For this, we use data from the International Stroke Trial (Sandercock et al., 2011), which is a randomized control trial (RCT) that examines the outcomes for early administration of aspirin, heparin, a combination of both, or none on acute ischaemic stroke. Importantly, this means that there are *four different* treatments available. The advantage of an RCT over observational data is that we can estimate the ground truth value function without bias, as the true propensity score is known. Our aim is to find the optimal treatment strategy based on patient covariates in order to prolong the *time-to-death* (TD) outcome variable (in days).

For this, we artificially introduce unobserved confounding as follows: In the training dataset, we randomly drop $60\%$ of the untreated patients whose diastolic blood pressure is larger than the average, as well as $60\%$ of the patients who received aspirin and whose blood pressure is lower than average. Then, we remove the diastolic blood pressure variable. Thereby, we introduce *unobserved* confounding in the training dataset.

**Results:** We report the estimated improvement of the TD outcome of all methods over the randomized policy in Figure 5. We report regret relative to a fully randomized (RCT) policy because, under unobserved confounding, it is the only baseline whose value remains identified and therefore provides a neutral, assumption-free reference point for evaluating policy improvement. Here, we vary the sensitivity parameter for both the method by Kallus & Zhou (2018a; 2021) and our method. Our method per-

Figure 5: **Real-world medical data.** We compare our efficient estimator against the previous baselines based on data from the International Stroke Trial. Our method yields the best treatment policy and is robust over different $\Gamma$.

forms best at $\Gamma = 24$. Further, our method has the overall best treatment strategy. Only for very small $\Gamma$, the performance of our method deteriorates – this is expected, as it does not guard against unobserved confounding and is therefore biased, and in contrast to standard methods, requires estimation of more complex nuisance functions. The baseline performance is generally poor. Specifically, we observe a similar pattern for the method by Kallus & Zhou (2018a; 2021) as in the robustness analysis; it only optimizes against a baseline policy, and quickly reverts to it as $\Gamma$ increases (see Supplement B). In contrast, our method is highly stable for appropriate parameterizations of $\Gamma$, which confirms the effectiveness of our method, and shows its applicability to medical scenarios.

**Conclusion:** We develop a novel semi-parametrically efficient estimator for sharp bounds on the value function under unobserved confounding. Our results provide a principled way for reliable decision-making from observational data, and show that robust policy learning can improve decision-making in sensitive applications such as healthcare.

## ACKNOWLEDGMENTS

This paper is supported by the DAAD program "Konrad Zuse Schools of Excellence in Artificial Intelligence", sponsored by the Federal Ministry of Education and Research.

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

## A  EXTENDED RELATED WORK

**Offline reinforcement learning under unobserved confounding:** Offline reinforcement learning deals with the problem of learning the optimal policy when the reward (value) function is defined over an infinite time horizon (Javurek et al., 2026). Therefore, these works rely upon techniques that are different from ours.

Some works focus on off-policy evaluation under unobserved confounding (Bennett et al., 2021; Kallus & Zhou, 2020) and even propose computationally efficient algorithms for this task (Kausik et al., 2024). However, these methods primarily focus on the identification of policy value bounds without semi-parametrically efficient estimation procedures. In contrast, Bennett et al. (2024) propose an efficient estimator for offline reinforcement learning. Different from our work, however, they require estimation of density ratios in order to evaluate the policy value. Further, Pace et al. (2024) propose a heuristic approach that learns representations of the unobserved confounders but does not provide theoretical guarantees for efficiency or unbiasedness. Shi et al. (2022) propose an approach that involves the approximation of bridge functions in partially observable Markov decision processes (POMDPs). Additionally, Shi et al. (2024) use mediators as auxiliary variables to construct confidence intervals for policy evaluation under unobserved confounding. Finally, Wang et al. (2021) improve sample efficiency in offline reinforcement learning under both observed and unobserved confounding.

Frauen et al. (2025b) develops a retargeting strategy for the conditional average treatment effect to balance estimation error and decision performance.

Further, Rambachan et al. (2025) study a slightly related but different problem setting: a *binary* selection decision $D \in \{0, 1\}$ with a *binary* selectively observed outcome $Y^*$, and they derive sharp bounds on *prediction and fairness* metrics (e.g. risk, ROC, precision–recall) under a *Mean Outcome Sensitivity Model (MOSM)*. The MOSM directly bounds the outcome gap

$$\delta(x) = P(Y^* = 1 \mid D = 0, X = x) - P(Y^* = 1 \mid D = 1, X = x), \tag{19}$$

and their sharpness guarantees hold relative to this MOSM.

Our work concerns *offline policy evaluation and learning* with (possibly) multi-valued actions $A \in \mathcal{A}$ and general outcomes $Y$, under a *Marginal Sensitivity Model (MSM)* on treatment assignment. We target the *policy value $V(\pi)$* and derive *closed-form, sharp MSM bounds* together with a semiparametrically efficient EIF-based estimator.

Importantly, Rambachan et al. (2025) show that propensity-based models such as the MSM *imply a particular MOSM* (via outcome bounds derived from Bayes' rule), but the resulting MOSM "may not exhaust all implications" of the original MSM. Thus MOSM bounds are generally *valid but not sharp* for the MSM. In contrast, our analysis works directly at the level of the MSM and characterizes the *sharp MSM-identified set* for $V(\pi)$.

# B    COMPARISON WITH KALLUS & ZHOU (2018A; 2021)

## B.1    TECHNICAL SUMMARY OF KALLUS & ZHOU (2018A; 2021)

Kallus & Zhou (2018a; 2021) work under the same marginal sensitivity model as we do, specified via an odds-ratio bound $\Gamma$ between the nominal propensity $e(A \mid X)$ and the true propensity $e(A \mid X, U)$. For a fixed policy $\pi$ and baseline policy $\pi_0$, they define an IPW/Hájek-type estimator for the *advantage* of $\pi$ over $\pi_0$, say $\lambda_i(\pi, \pi_0)$ for samples $i \in \{1, \ldots, n\}$, and consider the set of feasible weights $w = (w_1, \ldots, w_n)$ implied by the marginal sensitivity model:

$$U_\Gamma \;=\; \big\{ w : \underline{w}_i(\Gamma) \leq w_i \leq \overline{w}_i(\Gamma), \; \tfrac{1}{n} \sum_{i=1}^{n} w_i = 1 \big\}. \tag{20}$$

Given $U_\Gamma$, they define the worst-case regret of $\pi$ relative to $\pi_0$ as

$$R_\Gamma(\pi; \pi_0) \;=\; \sup_{w \in U_\Gamma} \frac{\sum_{i=1}^{n} w_i \, \lambda_i(\pi, \pi_0)}{\sum_{i=1}^{n} w_i}, \tag{21}$$

and then select the policy that *minimizes* this worst-case regret. In the binary-action implementation, the inner optimization is solved by sorting the $\lambda_i(\pi, \pi_0)$ and exploiting the fact that the linear–fractional program over $U_\Gamma$ attains its optimum at an extreme point: there exists a single index $k^* \in \{1, \ldots, n\}$ such that units on one side of $k^*$ are assigned their upper feasible weight and units on the other side their lower feasible weight. Hence the inner supremum reduces to a one-dimensional search over $i$ to find $k^*$, which Kallus & Zhou (2018a; 2021) implement via ternary search.

## B.2    OUR APPROACH AND KEY DIFFERENCES

Our work builds on the same marginal sensitivity model but differs in several technical aspects:

- *Objective: regret vs. value bounds.* Kallus & Zhou (2018a; 2021) optimize *worst-case regret* relative to a baseline policy, which is sharp for that regret quantity but does not provide closed-form sharp bounds on the *value* $V(\pi)$ itself. In contrast, we derive explicit *closed-form sharp upper and lower bounds* on $V(\pi)$ under the marginal sensitivity model and directly optimize these bounds over the policy class.

- *Propagation of IPW instability.* In Kallus & Zhou (2018a; 2021), the worst-case weights are determined by a single threshold $k^*$ on the sorted IPW/Hájek-based scores $\lambda_i$. Because $k^*$ is global, small estimation errors in the inverse propensities (and hence in the $\lambda_i$) can change which constraints bind in the inner problem and thus alter $k^*$. This then leads to a global change in the worst-case weights assigned to all units. Hence, local IPW noise can directly propagate into large changes in the estimated worst-case regret.

- *Closed-form sharp bounds and local error propagation.* Our estimator avoids this discrete minimax search and global thresholding. We derive sharp value bounds in closed form and estimate them with a semi-parametrically efficient, one-step bias-corrected estimator. That is, nuisance estimation errors then do **not** enter to the final bound estimate as first order bias, but only as lower order biases. Further, there is **no** single threshold index $k^*$ in our method whose mis-estimation can flip the worst-case assignment for the entire sample. This yields substantially more stable behavior in practice.

- *Sharpness and efficiency.* While Kallus & Zhou (2018a; 2021) show sharpness for the regret interval relative to a baseline, our results provide the *globally sharp identified set for $V(\pi)$ itself*. Moreover, our one-step estimator based on the EIF is semi-parametrically efficient and Neyman-orthogonal with respect to the nuisances, whereas the Hájek/IPW-type estimators used in (Kallus & Zhou, 2018a; 2021) do not achieve this efficiency bound.

- *Sharp regret does not imply sharp value.* The Kallus–Zhou method yields the sharp identified set for the *regret* $R(P) = V(\pi; P) - V(\pi_0; P)$ over $P \in \mathcal{P}(\Gamma)$, but this does **not** imply sharp bounds on $V(\pi)$ itself. The distributions that maximize regret and those that maximize the baseline value generally do not coincide. Consequently, the implied value upper bound satisfies only the loose inequality

$$\sup_{P \in \mathcal{P}(\Gamma)} V(\pi; P) \;<\; \sup_{P \in \mathcal{P}(\Gamma)} \big(V(\pi; P) - V(\pi_0; P)\big) \;+\; \sup_{P \in \mathcal{P}(\Gamma)} V(\pi_0; P), \qquad (22)$$

and this inequality is typically *strict*. Hence, even a sharp regret interval does not translate into the sharp identified set for $V(\pi)$. In contrast, our Theorem 4.1 directly characterizes the closed-form sharp bounds for $V(\pi)$ under the MSM.

Overall, we move from numerical minimax regret based on unstable IPW weights to closed-form sharp value bounds with an efficient, influence-function–based estimator that is both statistically and numerically more robust.

## C  ADDITIONAL RESULTS: COMPLEX PROPENSITY SCORES

In this section, we provide additional experimental results that illustrate the mechanism by which our efficient and sharp estimator improves over the method by Kallus & Zhou (2018a; 2021). In particular, we show how estimation errors in increasingly complex propensity score models propagate differently in the two approaches.

**Experimental setup:** We extend the synthetic data-generating process from Section 5 beyond binary actions and allow for

$$a \in \{0, 1, \ldots, d_a - 1\}. \tag{23}$$

We fix the unobserved confounding at $\Gamma^* = 5$ and run both methods with the same correctly specified sensitivity parameter $\Gamma = 5$. Increasing $d_a$ substantially complicates propensity esti-

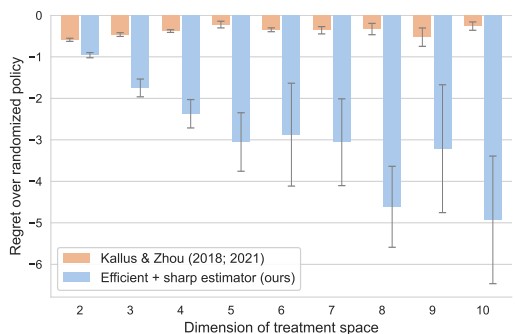

Figure 6: Performance comparison between our efficient and sharp method and Kallus & Zhou (2018a; 2021) under increasing dimension $d_a$ of the action-space.

mation: estimating $e(a \mid X)$ becomes a multiclass classification problem, and the resulting inverse-propensity terms become noisier as $d_a$ grows. This provides an ideal setting to study how each method handles nuisance complexity.

**Results:** Figure 6 summarizes the results. The findings align with our theoretical discussion in Supplement B:

- **Our method remains stable.** Our estimator maintains tight value bounds and stable performance as $d_a$ increases. The one-step correction based on the efficient influence function ensures that first-order sensitivity to errors in the estimated nuisances is eliminated. Consequently, even as the propensity scores become harder to estimate, the resulting bounds remain well behaved and lead to an increasingly better, confounding-robust policy.

- **Kallus & Zhou (2018a; 2021) become increasingly conservative.** In contrast, the approach of Kallus & Zhou (2018a; 2021) again reverts to the baseline policy. Their method computes worst-case regret relative to $\pi_0$ via a minimax optimization over feasible weights $U_\Gamma$:

$$R_\Gamma(\pi; \pi_0) = \sup_{w \in U_\Gamma} \frac{\sum_{i=1}^n w_i \lambda_i(\pi, \pi_0)}{\sum_{i=1}^n w_i},$$

where the $\lambda_i(\pi, \pi_0)$ are IPW/Hájek-type advantage scores. The optimizer selects a single global threshold $k^*$ (via ternary search) after sorting the $\lambda_i$ to determine which observations receive lower versus upper feasible weights.

When $d_a$ grows, the estimated $\lambda_i$ become highly variable due to propensity estimation noise. This causes the inner maximization to lock onto overly pessimistic weights in which many non-baseline units receive their lower feasible weight. As a result, nearly all non-baseline policies appear unsafe, and the learned policy collapses back to $\pi_0$, even though the true confounding level is correctly specified.

# D  IHDP DATASET

| | Γ = 1.0 | 1.1 | 1.2 | 1.3 | 1.4 | 1.5 | 1.6 | 1.7 | 1.8 | 1.9 | 2.0 | 2.1 | 2.2 | 2.3 |
|---|---|---|---|---|---|---|---|---|---|---|---|---|---|---|
| Standard IPW estimator | $-0.32 \pm 0.15$ | $-0.32 \pm 0.15$ | $-0.32 \pm 0.15$ | $-0.32 \pm 0.15$ | $-0.32 \pm 0.15$ | $-0.32 \pm 0.15$ | $-0.32 \pm 0.15$ | $-0.32 \pm 0.15$ | $-0.32 \pm 0.15$ | $-0.32 \pm 0.15$ | $-0.32 \pm 0.15$ | $-0.32 \pm 0.15$ | $-0.32 \pm 0.15$ | $-0.32 \pm 0.15$ |
| Standard DR estimator | $-2.19 \pm 0.41$ | $-2.19 \pm 0.41$ | $-2.19 \pm 0.41$ | $-2.19 \pm 0.41$ | $-2.19 \pm 0.41$ | $-2.19 \pm 0.41$ | $-2.19 \pm 0.41$ | $-2.19 \pm 0.41$ | $-2.19 \pm 0.41$ | $-2.19 \pm 0.41$ | $-2.19 \pm 0.41$ | $-2.19 \pm 0.41$ | $-2.19 \pm 0.41$ | $-2.19 \pm 0.41$ |
| Kallus & Zhou (2018a; 2021) | $-0.02 \pm 0.15$ | $-0.02 \pm 0.14$ | $-0.01 \pm 0.14$ | $-0.00 \pm 0.15$ | $-0.02 \pm 0.14$ | $0.02 \pm 0.13$ | $0.03 \pm 0.13$ | $0.02 \pm 0.13$ | $-0.01 \pm 0.13$ | $0.01 \pm 0.14$ | $0.02 \pm 0.12$ | $0.02 \pm 0.12$ | $0.01 \pm 0.12$ | $-0.01 \pm 0.13$ |
| **Efficient + sharp (ours)** | $\mathbf{-2.42 \pm 0.31}$ | $\mathbf{-2.46 \pm 0.32}$ | $\mathbf{-2.92 \pm 0.22}$ | $\mathbf{-2.68 \pm 0.27}$ | $\mathbf{-2.84 \pm 0.24}$ | $\mathbf{-2.88 \pm 0.26}$ | $\mathbf{-3.05 \pm 0.25}$ | $\mathbf{-2.89 \pm 0.27}$ | $\mathbf{-2.82 \pm 0.30}$ | $\mathbf{-2.65 \pm 0.28}$ | $\mathbf{-2.58 \pm 0.25}$ | $\mathbf{-2.57 \pm 0.25}$ | $\mathbf{-2.54 \pm 0.30}$ | $\mathbf{-2.60 \pm 0.31}$ |
| **Absolute improvement** | $-0.23$ | $-0.27$ | $-0.74$ | $-0.49$ | $-0.65$ | $-0.70$ | $-0.86$ | $-0.70$ | $-0.63$ | $-0.46$ | $-0.39$ | $-0.38$ | $-0.35$ | $-0.41$ |

Table 3: **Regret over randomized policy for varying sensitivity parameter** $\Gamma$. Lower values indicate better performance. Absolute improvement is computed against the best-performing baseline.

**Dataset:**  The IHDP dataset is a widely used semi-synthetic benchmark based on the Infant Health and Development Program (IHDP) randomized trial. Here, the effect of home visits on cognitive test scores for infants are measured. Covariates are taken from the real RCT, and potential outcomes are simulated following Hill (2011).

Similar to our real-world data study in Section 5, because the original study is unconfounded, we can introduce a known amount of unobserved confounding in a controlled way: here, we select covariates *birth weight (indicator of neonatal health)*, *head circumference at birth (developmental marker)*, and *weeks of gestation (prematurity measure)*, and drop $60\%$ of the observations whose value for any of the three covariates is below average and has received home visits, as well as $60\%$ of the observations whose value is below the average and has received *no* home visits. Then, we drop the three covariates and, thereby, introduce unobserved confounding. This mimics a realistic setting where key health indicators are absent from the recorded data but nevertheless influence treatment decisions and the outcome variable.

**Results:**  We evaluate the regret of all methods relative to a randomized policy because, under unobserved confounding, it is the only baseline whose value remains identified, whereas any non-randomized policy would implicitly rely on violated unconfoundedness assumptions and thus provide a misleading comparison.

The results in Table 3 closely mirror the patterns in our previous real–world experiment in Section 5. As expected, the standard IPW and DR baselines are biased and completely insensitive to the choice of $\Gamma$, since they assume unconfoundedness regardless of the true data-generating process.

The method of Kallus & Zhou (2018a; 2021) again behaves overly conservatively. In particular, it effectively reverts back toward its baseline policy and produces regret over the RCT baseline that cluster near zero. This, again, shows that the worst-case optimization collapses under even moderate confounding.

In contrast, our efficient and sharp estimator consistently achieves the lowest regret across all values of $\Gamma$. This demonstrates that our method computes meaningful bounds on the unidentified value function and, on top, finds optimizes toward the global best performing, confounding-robust policy.

# E    CHOOSING THE SENSITIVITY PARAMETER IN THE MSM

In this work, we adopt the MSM (Tan, 2006) in order to bound the ratio between the *nominal propensity score*

$$e(a, x) = p(A = a \mid X = x), \tag{24}$$

and the *true propensity score*

$$e(a, x, u) = p(A = a \mid X = x, U = u). \tag{25}$$

Here, the nominal propensity score can be estimated from data, whereas the true propensity score is fundamentally unobservable. In particular, the MSM is given by

$$\Gamma^{-1} \le \frac{e(a, x)}{1 - e(a, x)} \frac{1 - e(a, x, u)}{e(a, x, u)} \le \Gamma \tag{26}$$

for some sensitivity parameter $\Gamma \ge 1$.

Typically, the sensitivity constraints $\Gamma$ are chosen via domain knowledge (Frauen et al., 2023; Kallus et al., 2019) or data-driven heuristics (Hatt et al., 2022). For example, in practical applications, one typically has a benchmark variable (e.g., hours with sunlight) that is a known cause of the outcome (e.g., vitamin D deficiency), and one then wants to study how strong a confounder (e.g., other ecological activities such as nutrition) must be to explain away the effect of the benchmark variables. (Cinelli & Hazlett, 2020) term this the robustness value, which quantifies the strength of unobserved confounding needed to change conclusions.

Hence, to achieve this, a commonly used strategy for selecting $\Gamma$ is the following: We can search for the smallest $\Gamma$ such that the partially identified interval for the causal quantity of interest includes $0$. Then, we can interpret $\Gamma$ as a measure of the minimal deviation from unconfoundedness required to invalidate the effect of an intervention (Jesson et al., 2021; Jin et al., 2023).

# F PROOFS

## F.1 SHARP BOUND OF THE VALUE FUNCTION

**Proposition F.1.** *Let $Q^{\pm,*}(a, x)$ be the sharp upper/lower bound for the conditional average potential outcome, given our sensitivity constraints $\mathcal{P}(\Gamma)$. Then, sharp bounds for the value function $V(\pi)$ are given by*

$$V^{\pm,*}(\pi) = \int_{\mathcal{X}} \sum_a Q^{\pm,*}(a, x)\, \pi(a \mid x)\, \mathrm{d}p(x). \tag{27}$$

*Proof.* We provide the proof for the sharp upper bound $V^{+,*}(\pi)$. The lower bound follows completely analogously by swapping the signs and replacing the supremum with an infimum.

We start by noting that the upper bound on the value function depends on the set of admissible distributions $\mathcal{P}(\Gamma)$ induced by the sensitivity model, that is,

$$V^{+,*}(\pi) = V^{+,*}(\pi, \mathcal{P}(\Gamma)). \tag{28}$$

Hence, we can write

$$V^{+,*}(\pi) \tag{29}$$

$$= V^{+,*}(\pi, \mathcal{P}(\Gamma)) \tag{30}$$

$$= \sup_{\tilde{p} \in \mathcal{P}(\Gamma)} V(\pi, \tilde{p}) \tag{31}$$

$$= \sup_{\tilde{p} \in \mathcal{P}(\Gamma)} \int_{\mathcal{X}} \sum_a Q(a, x, \tilde{p})\, \pi(a \mid x)\, \mathrm{d}\tilde{p}(x) \tag{32}$$

$$= \sup_{\tilde{p} \in \mathcal{P}(\Gamma)} \int_{\mathcal{X}} \sum_a Q(a, x, \tilde{p})\, \pi(a \mid x)\, \mathrm{d}p(x), \tag{33}$$

where Equation 33 follows from the equality $p(\mathcal{D}) = \tilde{p}(\mathcal{D})$ for all $\tilde{p} \in \mathcal{P}(\Gamma)$.

Clearly, by definition of the optimal bounds $Q^{+,*}(a, x)$, we have that

$$Q(a, x, \tilde{p}) \leq \sup_{\tilde{p} \in \mathcal{P}(\Gamma)} Q(a, x, \tilde{p}) = Q^{+,*}(a, x, \mathcal{P}(\Gamma)) \tag{34}$$

for all $\tilde{p} \in \mathcal{P}(\Gamma)$, and since $Q^{+,*}(a, x) \in L^1(\pi, p)$, we know by dominated convergence (Frauen et al., 2023) that

$$\sup_{\tilde{p} \in \mathcal{P}(\Gamma)} \int_{\mathcal{X}} \sum_a Q(a, x, \tilde{p})\, \pi(a \mid x)\, \mathrm{d}p(x) = \int_{\mathcal{X}} \sum_a Q^{+,*}(a, x)\, \pi(a \mid x)\, \mathrm{d}p(x). \tag{35}$$

$\square$

### F.2 EFFICIENT ESTIMATOR OF THE SHARP BOUND OF THE VALUE FUNCTION

**On the construction of semi-parametrically efficient estimators.** Let $\Psi(P) = V^{+,*}(\pi; P)$ denote our target functional in the nonparametric model, and let $\mathcal{T}(P)$ be the corresponding tangent space. For any regular parametric submodel $\{P_\varepsilon\}$ through $P$ with score $s_\varepsilon(Z) = \partial_\varepsilon \log p_\varepsilon(Z)|_{\varepsilon=0}$, the pathwise derivative of $\Psi$ at $P$ in direction $s \in \mathcal{T}(P)$ is

$$\dot{\Psi}_P(s) := \left. \frac{d}{d\varepsilon} \Psi(P_\varepsilon) \right|_{\varepsilon=0}.$$

The *canonical gradient* $g^\star \in L_0^2(P)$ is defined as the unique element such that

$$\dot{\Psi}_P(s) = \mathbb{E}_P\big[g^\star(Z)\, s(Z)\big] \quad \text{for all } s \in \mathcal{T}(P), \tag{36}$$

i.e., $g^\star$ is the Riesz representer of the pathwise derivative functional (see, e.g., van der Vaart (1998), Chapter 25).

In the following, we derive the influence function of $V^{+,*}(\pi)$ via the chain rule for pathwise differentiability. Specifically, we decompose $V^{+,*}(\pi)$ into building blocks (conditional expectations, conditional distributions), plug in the known canonical gradients for each of these primitives, and then apply the chain rule as in Kennedy (2022) and Lemma S3 of Luedtke (2024). By construction, the resulting function $\varphi(Z)$ satisfies

$$\left. \frac{d}{d\varepsilon} \Psi(P_\varepsilon) \right|_{\varepsilon=0} = \mathbb{E}_P\big[\varphi(Z)\, s_\varepsilon(Z)\big] \quad \text{for every regular submodel } \{P_\varepsilon\}. \tag{37}$$

Hence $\varphi$ is exactly the canonical gradient $g^\star$, and therefore the efficient influence function in the nonparametric model. By standard semiparametric theory (e.g., van der Vaart (1998)), the one-step estimator based on this $\varphi$ is semiparametrically efficient for $V^{+,*}(\pi)$.

**Theorem F.2.** *Let $\mathbb{E}[|Y|] < \infty$ and $p(y \mid x, a)$ admit a continuous density bounded away from zero in a neighborhood of $F_{x,a}^{-1}(\alpha^+)$. This holds, for example, if the density $p(y \mid x, a)$ is strictly positive and continuous. Then, an estimator for the sharp upper bound of the value function is given by*

$$
\begin{aligned}
\hat{V}^{+,*}(\pi) = \mathbb{P}_n\Big\{ &\sum_a \pi_{a,X}\Big[\hat{Q}_{a,X}^{+,*} - \hat{e}_{a,X}\Big(b^-\underline{\hat{\mu}}_{a,X}^+ + b^+\hat{\bar{\mu}}_{a,X}^+\Big)\Big] + \pi_{A,X}\Big(b^-\underline{\hat{\mu}}_{A,X}^+ + b^+\hat{\bar{\mu}}_{A,X}^+\Big) \\
&+ \frac{\pi_{A,X}}{\hat{e}_{A,X}}\Big[\Big(\hat{c}_{A,X}^- - \hat{c}_{A,X}^+\Big)\Big(\hat{F}_{X,A}^{-1}(\alpha^+)(\underline{\hat{\Delta}}_{Y,A,X}^+ - \alpha^+)\Big) \\
&+ \hat{c}_{A,X}^-\Big(Y\underline{\hat{\Delta}}_{Y,A,X}^+ - \underline{\hat{\mu}}_{A,X}^+\Big) + \hat{c}_{A,X}^+\Big(Y\hat{\bar{\Delta}}_{Y,A,X}^+ - \hat{\bar{\mu}}_{A,X}^+\Big)\Big]\Big\}.
\end{aligned}
\tag{38}
$$

*If, on top, $\mathbb{E}[|Y|^2] < \infty$, the above estimator is* **semi-parametrically efficient**.

*Proof.* The sharp upper bound of the value function is given by

$$V^{\pm,*}(\pi) = \int_{\mathcal{X}} \sum_a Q^{\pm,*}(a, x)\pi(a \mid x)\, \mathrm{d}p(x). \tag{39}$$

In the following, we derive the efficient estimator for this quantity. Therein, we make use of the chain rule for deriving efficient influence function (Kennedy, 2022). *A proof of the validity of the chain rule for deriving efficient influence functions is provided by Luedtke (2024)* (**Lemma S3**).

In order to avoid notational overload and for the sake of clarity, we do not use additional variables such as $b^\pm, c^\pm, \underline{\Delta}^\pm, \bar{\mu}^\pm$, etc. until the final steps, such that the derivation becomes easier to follow. Moreover, we make the dependency on nuisance functions $\eta \subseteq \{e(a, x), F_{a,x}^{-1}(\alpha^\pm), \bar{\mu}^\pm(a, x), \underline{\mu}^\pm(a, x)\}$ explicit by writing, for example, $V^{+,*}(\pi; \eta)$ for $V^{+,*}(\pi)$.

The influence function of $V^{+,*}(\pi; \eta)$ is given by

$$\mathbb{IF}\left(V^{+,*}(\pi; \eta)\right) \tag{40}$$

$$=\mathbb{IF}\left(\int_{\mathcal{X}} \sum_a Q^{+,*}(a, x; \eta)\pi(a \mid x)\, \mathrm{d}p(x)\right) \tag{41}$$

$$=\sum_a \int_{\mathcal{X}} \pi(a \mid x)\, \mathbb{IF}\left(p(x)Q^{+,*}(a, x; \eta)\right)\, \mathrm{d}x \tag{42}$$

$$=\sum_a \int_{\mathcal{X}} \pi(a \mid x)\, \mathbb{IF}\left(p(x)\right)Q^{+,*}(a, x; \eta) + \pi(a \mid x)\, p(x)\, \mathbb{IF}\left(Q^{+,*}(a, x; \eta)\right)\, \mathrm{d}x \tag{43}$$

$$=\sum_a \int_{\mathcal{X}} \pi(a \mid x)\left(\mathbb{1}_{\{X=x\}} - p(x)\right)Q^{+,*}(a, x; \eta)\, \mathrm{d}x + \sum_a \int_{\mathcal{X}} \pi(a \mid x)\, p(x)\, \mathbb{IF}\left(Q^{+,*}(a, x; \eta)\right)\, \mathrm{d}x \tag{44}$$

$$=\sum_a \pi(a \mid X)\, Q^{+,*}(a, X; \eta) - V^{+,*}(\pi; \eta) + \sum_a \int_{\mathcal{X}} \pi(a \mid x)\, p(x)\, \mathbb{IF}\left(Q^{+,*}(a, x; \eta)\right)\, \mathrm{d}x \tag{45}$$

Hence, in Equation 45, we are left to compute the efficient influence function of $Q^{+,*}(a, x)$, that is, the sharp upper bound of the CAPO. With $\alpha^+ = \Gamma/(1 + \Gamma)$, the sharp upper bound $Q^{+,*}(a, x)$ is given by

$$Q^{+,*}(a, x; \eta) \tag{46}$$

$$=\left((1 - \Gamma^{-1})e(a, x) + \Gamma^{-1}\right)\mathbb{E}\left[Y\mathbb{1}_{\{Y \leq F_{X,A}^{-1}(\alpha^+)\}} \mid X = x, A = a\right] \tag{47}$$

$$+ \left((1 - \Gamma)e(a, x) + \Gamma\right)\mathbb{E}\left[Y\mathbb{1}_{\{Y \geq F_{X,A}^{-1}(\alpha^+)\}} \mid X = x, A = a\right]. \tag{48}$$

Hence, the influence function is given by

$$\mathbb{IF}\left(Q^{+,*}(a, x; \eta)\right) \tag{49}$$

$$=\underbrace{\mathbb{IF}\left((1 - \Gamma^{-1})e(a, x) + \Gamma^{-1}\right)}_{(a)} \mathbb{E}\left[Y\mathbb{1}_{\{Y \leq F_{X,A}^{-1}(\alpha^+)\}} \mid X = x, A = a\right] \tag{50}$$

$$+ \left((1 - \Gamma^{-1})e(a, x) + \Gamma^{-1}\right)\underbrace{\mathbb{IF}\left(\mathbb{E}\left[Y\mathbb{1}_{\{Y \leq F_{X,A}^{-1}(\alpha^+)\}} \mid X = x, A = a\right]\right)}_{(c)} \tag{51}$$

$$+ \underbrace{\mathbb{IF}\left((1 - \Gamma)e(a, x) + \Gamma\right)}_{(b)} \mathbb{E}\left[Y\mathbb{1}_{\{Y \geq F_{X,A}^{-1}(\alpha^+)\}} \mid X = x, A = a\right] \tag{52}$$

$$+ \left((1 - \Gamma)e(a, x) + \Gamma\right)\underbrace{\mathbb{IF}\left(\mathbb{E}\left[Y\mathbb{1}_{\{Y \geq F_{X,A}^{-1}(\alpha^+)\}} \mid X = x, A = a\right]\right)}_{(d)}, \tag{53}$$

where we can apply the chain rule to the conditional expectation of the truncated first moments of $Y$, since $Y$ is integrable by assumption. We start with $(a)$ and $(b)$. For $(a)$, we obtain

$$\mathbb{IF}\left((1 - \Gamma^{-1})e(a, x) + \Gamma^{-1}\right) \tag{54}$$

$$=(1 - \Gamma^{-1})\mathbb{IF}\left(e(a, x)\right) \tag{55}$$

$$=(1 - \Gamma^{-1})\frac{\mathbb{1}_{\{X=x\}}}{p(x)}\left(\mathbb{1}_{\{A=a\}} - e(a, x)\right), \tag{56}$$

and, similarly for $(b)$, we yield

$$\mathbb{IF}\left((1 - \Gamma)e(a, x) + \Gamma\right) \tag{57}$$

$$=(1 - \Gamma)\frac{\mathbb{1}_{\{X=x\}}}{p(x)}\left(\mathbb{1}_{\{A=a\}} - e(a, x)\right). \tag{58}$$

Next, we compute the influence function of $(c)$ via

$$\mathbb{IF}\left(\mathbb{E}\left[Y\mathbb{1}_{\{Y\leq F_{X,A}^{-1}(\alpha^+)\}} \mid X = x, A = a\right]\right) \tag{59}$$

$$=\mathbb{IF}\left(\int_{\mathcal{Y}} \mathbb{1}_{\{y\leq F_{x,a}^{-1}(\alpha^+)\}} \, y \, p(y \mid x, a) \, \mathrm{d}y\right) \tag{60}$$

$$= \int_{\mathcal{Y}} \mathbb{IF}\left(\mathbb{1}_{\{y\leq F_{x,a}^{-1}(\alpha^+)\}}\right) y \, p(y \mid x, a) + \mathbb{1}_{\{y\leq F_{x,a}^{-1}(\alpha^+)\}} y \mathbb{IF}\left(p(y \mid x, a)\right) \mathrm{d}y \tag{61}$$

$$= \underbrace{\int_{\mathcal{Y}} \mathbb{IF}\left(\mathbb{1}_{\{y\leq F_{x,a}^{-1}(\alpha^+)\}}\right) y \, p(y \mid x, a) \, \mathrm{d}y}_{(c_1)} + \underbrace{\int_{\mathcal{Y}} \mathbb{1}_{\{y\leq F_{x,a}^{-1}(\alpha^+)\}} \, y \, \mathbb{IF}\left(p(y \mid x, a)\right) \mathrm{d}y}_{(c_2)}. \tag{62}$$

For $(c_1)$, we first note that

$$\mathbb{IF}\left(F_{x,a}(\alpha^+)\right) \tag{63}$$

$$=\mathbb{IF}\left(\mathbb{P}(Y \leq y \mid X = x, A = a)\right) \tag{64}$$

$$=\mathbb{IF}\left(\mathbb{E}[\mathbb{1}_{\{Y\leq y\}} \mid X = x, A = a]\right) \tag{65}$$

$$=\frac{\mathbb{1}_{\{X=x,A=a\}}}{p(a,x)}\left(\mathbb{1}_{\{Y\leq y\}} - \mathbb{E}[\mathbb{1}_{\{Y\leq y\}} \mid X = x, A = a]\right) \tag{66}$$

$$= \underbrace{\frac{\mathbb{1}_{\{X=x,A=a\}}}{p(a,x)}\left(\mathbb{1}_{\{Y\leq y\}} - F_{x,a}(y)\right)}_{(*)}. \tag{67}$$

Then, we can simplify $(c_1)$ via

$$\int_{\mathcal{Y}} \mathbb{IF}\left(\mathbb{1}_{\{y\leq F_{x,a}^{-1}(\alpha^+)\}}\right) y \, p(y \mid x, a) \, \mathrm{d}y \tag{68}$$

$$= \int_{\mathcal{Y}} \delta\left(y - F_{x,a}^{-1}(\alpha^+)\right)\mathbb{IF}\left(F_{x,a}^{-1}(\alpha^+)\right) y \, p(y \mid x, a) \, \mathrm{d}y \tag{69}$$

$$=\mathbb{IF}\left(F_{x,a}^{-1}(\alpha^+)\right) \int_{\mathcal{Y}} \delta\left(y - F_{x,a}^{-1}(\alpha^+)\right) y \, p(y \mid x, a) \, \mathrm{d}y \tag{70}$$

$$=\mathbb{IF}\left(F_{x,a}^{-1}(F_{x,a}(y^*))\right) \int_{\mathcal{Y}} \delta\left(y - F_{x,a}^{-1}(\alpha^+)\right) y \, p(y \mid x, a) \, \mathrm{d}y \tag{71}$$

$$=\frac{\mathrm{d}}{\mathrm{d}q}F_{x,a}^{-1}(q)\Big|_{q=F_{x,a}(y^*)} \mathbb{IF}\left(F_{x,a}(y^*)\right) \int_{\mathcal{Y}} \delta\left(y - F_{x,a}^{-1}(\alpha^+)\right) y \, p(y \mid x, a) \, \mathrm{d}y \tag{72}$$

$$=\frac{1}{F_{x,a}'(F_{x,a}^{-1}(F_{x,a}(y^*)))} \mathbb{IF}\left(F_{x,a}(y^*)\right) \int_{\mathcal{Y}} \delta\left(y - F_{x,a}^{-1}(\alpha^+)\right) y \, p(y \mid x, a) \, \mathrm{d}y \tag{73}$$

$$=\frac{1}{p(y^* \mid x, a)} \mathbb{IF}\left(F_{x,a}(y^*)\right) \int_{\mathcal{Y}} \delta\left(y - F_{x,a}^{-1}(\alpha^+)\right) y \, p(y \mid x, a) \, \mathrm{d}y \tag{74}$$

$$\overset{(*)}{=}\frac{1}{p(y^* \mid x, a)}\frac{\mathbb{1}_{\{X=x,A=a\}}}{p(a,x)}\left(\mathbb{1}_{\{Y\leq y^*\}} - F_{x,a}(y^*)\right) \int_{\mathcal{Y}} \delta(y - y^*) \, y \, p(y \mid x, a) \, \mathrm{d}y \tag{75}$$

$$=\frac{1}{p(y^* \mid x, a)}\frac{\mathbb{1}_{\{X=x,A=a\}}}{p(a,x)}\left(\mathbb{1}_{\{Y\leq y^*\}} - \alpha^+\right) \int_{\mathcal{Y}} \delta(y - y^*) \, y \, p(y \mid x, a) \, \mathrm{d}y \tag{76}$$

$$=\frac{1}{p(y^* \mid x, a)}\frac{\mathbb{1}_{\{X=x,A=a\}}}{p(a,x)}\left(\mathbb{1}_{\{Y\leq y^*\}} - \alpha^+\right) y^* \, p(y^* \mid x, a) \tag{77}$$

$$=\frac{\mathbb{1}_{\{X=x,A=a\}}}{p(a,x)}F_{x,a}^{-1}(\alpha^+)(\mathbb{1}_{\{Y\leq F_{x,a}^{-1}(\alpha^+)\}} - \alpha^+), \tag{78}$$

for some $y^* \in \mathcal{Y}$ such that $F_{x,a}(y^*) = \alpha^+$, where $\delta(\cdot)$ is the Dirac-delta function, and where we use in Equation 73 that $p(y|x,a)$ is a continuous density strictly bounded away from zero in a

neighborhood of $F_{x,a}^{-1}(\alpha^+)$, which guarantees that the *conditional quantile is unique*, and where Equation 74 *exists* by positivity of the conditional density.

Next, we simplify $(c_2)$ via

$$\int_{\mathcal{Y}} \mathbb{1}_{\{y \leq F_{x,a}^{-1}(\alpha^+)\}} y \mathbb{IF}\Big(p(y \mid x, a)\Big) \, \mathrm{d}y \tag{79}$$

$$= \int_{\mathcal{Y}} \mathbb{1}_{\{y \leq F_{x,a}^{-1}(\alpha^+)\}} \, y \, \mathbb{IF}\Big(\mathbb{E}[\mathbb{1}_{\{Y=y\}} \mid X = x, A = a]\Big) \, \mathrm{d}y \tag{80}$$

$$= \int_{\mathcal{Y}} \mathbb{1}_{\{y \leq F_{x,a}^{-1}(\alpha^+)\}} \, y \, \frac{\mathbb{1}_{\{X=x,A=a\}}}{p(a,x)} \Big(\mathbb{1}_{\{Y=y\}} - p(y \mid x, a)\Big) \, \mathrm{d}y \tag{81}$$

$$= \frac{\mathbb{1}_{\{X=x,A=a\}}}{p(a,x)} \Big(Y \mathbb{1}_{\{Y \leq F_{x,a}^{-1}(\alpha^+)\}} - \mathbb{E}[Y \mathbb{1}_{\{Y \leq F_{X,A}^{-1}(\alpha^+)\}} \mid X = x, A = a]\Big). \tag{82}$$

Then, combining $(c_1)$ and $(c_2)$, we get

$$\mathbb{IF}\Big(\mathbb{E}\Big[Y \mathbb{1}_{\{Y \leq F_{X,A}^{-1}(\alpha^+)\}} \mid X = x, A = a\Big]\Big) \tag{83}$$

$$= \frac{\mathbb{1}_{\{X=x,A=a\}}}{p(a,x)} \Big(Y \mathbb{1}_{\{Y \leq F_{x,a}^{-1}(\alpha^+)\}} - \mathbb{E}[Y \mathbb{1}_{\{Y \leq F_{x,a}^{-1}(\alpha^+)\}} \mid X = x, A = a] + F_{x,a}^{-1}(\alpha^+)(\mathbb{1}_{\{Y \leq F_{x,a}^{-1}(\alpha^+)\}} - \alpha^+)\Big). \tag{84}$$

Finally, we compute the influence function of $(d)$ analogously to $(c)$ via

$$\mathbb{IF}\Big(\mathbb{E}\Big[Y \mathbb{1}_{\{Y \geq F_{X,A}^{-1}(\alpha^+)\}} \mid X = x, A = a\Big]\Big) \tag{85}$$

$$= \underbrace{\int_{\mathcal{Y}} \mathbb{IF}\Big(\mathbb{1}_{\{y \geq F_{x,a}^{-1}(\alpha^+)\}}\Big) y \, p(y \mid x, a) \, \mathrm{d}y}_{(d_1)} + \underbrace{\int_{\mathcal{Y}} \mathbb{1}_{\{y \geq F_{x,a}^{-1}(\alpha^+)\}} y \, \mathbb{IF}\Big(p(y \mid x, a)\Big) \, \mathrm{d}y}_{(d_2)}. \tag{86}$$

Again, we start with $(d_1)$ via

$$\int_{\mathcal{Y}} \mathbb{IF}\Big(\mathbb{1}_{\{y \geq F_{x,a}^{-1}(\alpha^+)\}}\Big) y \, p(y \mid x, a) \, \mathrm{d}y \tag{87}$$

$$= \int_{\mathcal{Y}} \mathbb{IF}\Big((1 - \mathbb{1}_{\{y \leq F_{x,a}^{-1}(\alpha^+)\}})\Big) y \, p(y \mid x, a) \, \mathrm{d}y \tag{88}$$

$$= - \int_{\mathcal{Y}} \mathbb{IF}\Big(\mathbb{1}_{\{y \leq F_{x,a}^{-1}(\alpha^+)\}}\Big) y \, p(y \mid x, a) \, \mathrm{d}y \tag{89}$$

$$= \frac{\mathbb{1}_{\{X=x,A=a\}}}{p(a,x)} \Big(-F_{x,a}^{-1}(\alpha^+)\Big)(\mathbb{1}_{\{Y \leq F_{x,a}^{-1}(\alpha^+)\}} - \alpha^+), \tag{90}$$

using the result for $(c_1)$ in Equation 78. Further, for $(d_2)$, we get that

$$\int_{\mathcal{Y}} \mathbb{1}_{\{y \geq F_{x,a}^{-1}(\alpha^+)\}} y \, \mathbb{IF}\Big(p(y \mid x, a)\Big) \, \mathrm{d}y \tag{91}$$

$$= \frac{\mathbb{1}_{\{X=x,A=a\}}}{p(a,x)} \Big(Y \mathbb{1}_{\{Y \geq F_{x,a}^{-1}(\alpha^+)\}} - \mathbb{E}[Y \mathbb{1}_{\{Y \geq F_{X,A}^{-1}(\alpha^+)\}} \mid X = x, A = a]\Big). \tag{92}$$

Combining $(d_1)$ and $(d_2)$, we obtain that

$$\mathbb{IF}\Big(\mathbb{E}\Big[Y \mathbb{1}_{\{Y \geq F_{x,a}^{-1}(\alpha^+)\}} \mid X = x, A = a\Big]\Big) \tag{93}$$

$$= \frac{\mathbb{1}_{\{X=x,A=a\}}}{p(a,x)} \Big(Y \mathbb{1}_{\{Y \geq F_{x,a}^{-1}(\alpha^+)\}} - \mathbb{E}[Y \mathbb{1}_{\{Y \geq F_{X,A}^{-1}(\alpha^+)\}} \mid X = x, A = a] - F_{x,a}^{-1}(\alpha^+)(\mathbb{1}_{\{Y \leq F_{x,a}^{-1}(\alpha^+)\}} - \alpha^+)\Big). \tag{94}$$

Finally, we can state the influence function of $Q^{+,*}(a,x)$ by combining $(a)$–$(d)$ in Equation 56, Equation 58, Equation 84, and Equation 94. We get

$$\mathbb{IF}\Big(Q^{+,*}(a,x;\eta)\Big) \tag{95}$$

$$=\frac{\mathbb{1}_{\{X=x\}}}{p(x)}(1-\Gamma^{-1})\Big(\mathbb{1}_{\{A=a\}}-e(a,x)\Big)\mathbb{E}\Big[Y\mathbb{1}_{\{Y\leq F_{X,A}^{-1}(\alpha^+)\}}\mid X=x,A=a\Big] \tag{96}$$

$$+\frac{\mathbb{1}_{\{X=x,A=a\}}}{p(a,x)}\Big((1-\Gamma^{-1})e(a,x)+\Gamma^{-1}\Big) \tag{97}$$

$$\times\Big(Y\mathbb{1}_{\{Y\leq F_{x,a}^{-1}(\alpha^+)\}}-\mathbb{E}[Y\mathbb{1}_{\{Y\leq F_{X,A}^{-1}(\alpha^+)\}}\mid X=x,A=a]+F_{x,a}^{-1}(\alpha^+)(\mathbb{1}_{\{Y\leq F_{x,a}^{-1}(\alpha^+)\}}-\alpha^+)\Big) \tag{98}$$

$$+\frac{\mathbb{1}_{\{X=x\}}}{p(x)}(1-\Gamma)\Big(\mathbb{1}_{\{A=a\}}-e(a,x)\Big)\mathbb{E}\Big[Y\mathbb{1}_{\{Y\geq F_{X,A}^{-1}(\alpha^+)\}}\mid X=x,A=a\Big] \tag{99}$$

$$+\frac{\mathbb{1}_{\{X=x,A=a\}}}{p(a,x)}\Big((1-\Gamma)e(a,x)+\Gamma\Big) \tag{100}$$

$$\times\Big(Y\mathbb{1}_{\{Y\geq F_{x,a}^{-1}(\alpha^+)\}}-\mathbb{E}[Y\mathbb{1}_{\{Y\geq F_{X,A}^{-1}(\alpha^+)\}}\mid X=x,A=a]-F_{x,a}^{-1}(\alpha^+)(\mathbb{1}_{\{Y\leq F_{x,a}^{-1}(\alpha^+)\}}-\alpha^+)\Big). \tag{101}$$

In order to simplify the above lengthy equation as in our main paper, we introduce the following variables:

- $b^{\pm}=(1-\Gamma^{\pm 1})$
- $c^{\pm}(a,x;\eta)=(b^{\pm}e(a,x)+\Gamma^{\pm 1})$
- $\underline{\Delta}^{\pm}(y,a,x;\eta)=\mathbb{1}_{\{y\leq F_{x,a}^{-1}(\alpha^{\pm})\}}$
- $\bar{\Delta}^{\pm}(y,a,x;\eta)=\mathbb{1}_{\{y\geq F_{x,a}^{-1}(\alpha^{\pm})\}}$
- $\underline{\mu}^{\pm}(a,x;\eta)=\mathbb{E}[Y\underline{\Delta}^{\pm}(Y,A,X;\eta)\mid X=x,A=a]$
- $\bar{\mu}^{\pm}(a,x;\eta)=\mathbb{E}[Y\bar{\Delta}^{\pm}(Y,A,X;\eta)\mid X=x,A=a]$

Then, Equation 101 simplifies to

$$\mathbb{IF}\Big(Q^{+,*}(a,x;\eta)\Big) \tag{102}$$

$$=\frac{\mathbb{1}_{\{X=x\}}}{p(x)}b^{-}\Big(\mathbb{1}_{\{A=a\}}-e(a,x)\Big)\underline{\mu}^{+}(a,x;\eta) \tag{103}$$

$$+\frac{\mathbb{1}_{\{X=x,A=a\}}}{p(x)e(a,x)}c^{-}(a,x;\eta)\Big(Y\underline{\Delta}^{+}(Y,a,x;\eta)-\underline{\mu}^{+}(a,x;\eta)+F_{x,a}^{-1}(\alpha^+)(\underline{\Delta}^{+}(Y,a,x;\eta)-\alpha^+)\Big) \tag{104}$$

$$+\frac{\mathbb{1}_{\{X=x\}}}{p(x)}b^{+}\Big(\mathbb{1}_{\{A=a\}}-e(a,x)\Big)\bar{\mu}^{+}(a,x;\eta) \tag{105}$$

$$+\frac{\mathbb{1}_{\{X=x,A=a\}}}{p(x)e(a,x)}c^{+}(a,x;\eta)\Big(Y\bar{\Delta}^{+}(Y,a,x;\eta)-\bar{\mu}^{+}(a,x;\eta)-F_{x,a}^{-1}(\alpha^+)(\underline{\Delta}^{+}(Y,a,x;\eta)-\alpha^+)\Big) \tag{106}$$

$$=\frac{\mathbb{1}_{\{X=x\}}}{p(x)}\Big\{\Big(\mathbb{1}_{\{A=a\}}-e(a,x)\Big)\Big(b^{-}\underline{\mu}^{+}(a,x;\eta)+b^{+}\bar{\mu}^{+}(a,x;\eta)\Big) \tag{107}$$

$$+\frac{\mathbb{1}_{\{A=a\}}}{e(a,x)}\Big[\Big(c^{-}(a,x;\eta)-c^{+}(a,x;\eta)\Big)\Big(F_{x,a}^{-1}(\alpha^+)(\underline{\Delta}^{+}(Y,a,x;\eta)-\alpha^+)\Big) \tag{108}$$

$$+c^{-}(a,x;\eta)\Big(Y\underline{\Delta}^{+}(Y,a,x;\eta)-\underline{\mu}^{+}(a,x;\eta)\Big)+c^{+}(a,x;\eta)\Big(Y\bar{\Delta}^{+}(Y,a,x;\eta)-\bar{\mu}^{+}(a,x;\eta)\Big)\Big]\Big\} \tag{109}$$

Finally, we can combine Equation 45 and Equation 109. That is, the efficient influence function of $V^{+,*}(\pi)$ is given by

$$\mathbb{IF}\Big(V^{+,*}(\pi;\eta)\Big) \tag{110}$$

$$=\sum_a \pi(a\mid X)Q^{+,*}(a,X) - V^{+,*}(\pi) \tag{111}$$

$$+\sum_a \pi(a\mid X)\Big(\mathbb{1}_{\{A=a\}} - e(a,X)\Big)\Big(b^-\underline{\mu}^+(a,X;\eta) + b^+\bar{\mu}^+(a,X;\eta)\Big) \tag{112}$$

$$+\sum_a \pi(a\mid X)\frac{\mathbb{1}_{\{A=a\}}}{e(a,X)}\Big[\Big(c^-(a,X;\eta) - c^+(a,X;\eta)\Big)\Big(F_{x,a}^{-1}(\alpha^+)(\underline{\Delta}^+(Y,a,X;\eta) - \alpha^+)\Big) \tag{113}$$

$$+ c^-(a,X;\eta)\Big(Y\underline{\Delta}^+(Y,a,X;\eta) - \underline{\mu}^+(a,X;\eta)\Big) + c^+(a,X;\eta)\Big(Y\bar{\Delta}^+(Y,a,X;\eta) - \bar{\mu}^+(a,X;\eta)\Big)\Big] \tag{114}$$

$$=-V^{+,*}(\pi) + \sum_a \pi(a\mid X)\Big[Q^{+,*}(a,X) - e(a,X)\Big(b^-\underline{\mu}^+(a,X;\eta) + b^+\bar{\mu}^+(a,X;\eta)\Big)\Big] \tag{115}$$

$$+\pi(A\mid X)\Big(b^-\underline{\mu}^+(A,X;\eta) + b^+\bar{\mu}^+(A,X;\eta)\Big) \tag{116}$$

$$+\frac{\pi(A\mid X)}{e(A,X)}\Big[\Big(c^-(A,X;\eta) - c^+(A,X;\eta)\Big)\Big(F_{X,A}^{-1}(\alpha^+)(\underline{\Delta}_{\alpha^+}(Y,A,X;\eta) - \alpha^+)\Big) \tag{117}$$

$$+ c^-(A,X;\eta)\Big(Y\underline{\Delta}^+(Y,A,X;\eta) - \underline{\mu}^+(A,X;\eta)\Big) + c^+(A,X;\eta)\Big(Y\bar{\Delta}^+(Y,A,X;\eta) - \bar{\mu}^+(A,X;\eta)\Big)\Big] \tag{118}$$

We can derive the efficient estimator for the bounds of the value function through one-step bias correction via

$$V^{+,*}(\pi;\hat{\eta}) + \mathbb{P}_n\Big\{V^{+,*}(\pi;\hat{\eta})\Big\} \tag{119}$$

$$=\mathbb{P}_n\Big\{\sum_a \pi(a\mid X)\Big[Q^{+,*}(a,X;\hat{\eta}) - \hat{e}(a,X)\Big(b^-\hat{\underline{\mu}}^+(a,X;\hat{\eta}) + b^+\hat{\bar{\mu}}^+(a,X;\hat{\eta})\Big)\Big] \tag{120}$$

$$+\pi(A\mid X)\Big(b^-\hat{\underline{\mu}}^+(A,X;\hat{\eta}) + b^+\hat{\bar{\mu}}^+(A,X;\hat{\eta})\Big) \tag{121}$$

$$+\frac{\pi(A\mid X)}{\hat{e}(A,X)}\Big[\Big(c^-(A,X;\hat{\eta}) - c^+(A,X;\hat{\eta})\Big)\Big(\hat{F}_{X,A}^{-1}(\alpha^+)(\underline{\Delta}^+(Y,A,X;\hat{\eta}) - \alpha^+)\Big) \tag{122}$$

$$+ c^-(A,X;\hat{\eta})\Big(Y\underline{\Delta}^+(Y,A,X;\hat{\eta}) - \hat{\underline{\mu}}^+(A,X;\hat{\eta})\Big) + c^+(A,X;\hat{\eta})\Big(Y\bar{\Delta}^+(Y,A,X;\hat{\eta}) - \hat{\bar{\mu}}^+(A,X;\hat{\eta})\Big)\Big]\Big\} \tag{123}$$

$$=\mathbb{P}_n\Big\{\sum_a \pi_{a,X}\Big[\hat{Q}_{a,X}^{+,*} - \hat{e}_{a,X}\Big(b^-\hat{\underline{\mu}}_{a,X}^+ + b^+\hat{\bar{\mu}}_{a,X}^+\Big)\Big] + \pi_{A,X}\Big(b^-\hat{\underline{\mu}}_{A,X}^+ + b^+\hat{\bar{\mu}}_{A,X}^+\Big) \tag{124}$$

$$+\frac{\pi_{A,X}}{\hat{e}_{A,X}}\Big[\Big(\hat{c}_{A,X}^- - \hat{c}_{A,X}^+\Big)\Big(\hat{F}_{X,A}^{-1}(\alpha^+)(\hat{\underline{\Delta}}_{Y,A,X}^+ - \alpha^+)\Big) + \hat{c}_{A,X}^-\Big(Y\hat{\underline{\Delta}}_{Y,A,X}^+ - \hat{\underline{\mu}}_{A,X}^+\Big) + \hat{c}_{A,X}^+\Big(Y\hat{\bar{\Delta}}_{Y,A,X}^+ - \hat{\bar{\mu}}_{A,X}^+\Big)\Big]\Big\}$$

using our short-hand notation from the main paper.

$\square$

## F.3 LEARNING GUARANTEE: VALUE FUNCTION

**Theorem F.3.** *Assume* $|Y| \le C_y$. *Let* $C_v := 2C_y(1 + \Gamma^{-1} + \Gamma)$ *and let* $R_n(\Pi)$ *denote the empirical Rademacher complexity of the policy class* $\Pi$. *Then, with probability at least* $1 - \delta$,

$$\sup_{\pi \in \Pi} \left\{ V(\pi) - \hat{V}^{+,*}(\pi) \right\} \le 2C_v \left( R_n(\Pi) + \tfrac{5}{2}\sqrt{\tfrac{1}{2n}\log\tfrac{2}{\delta}} \right). \tag{125}$$

*Equivalently, on the same high-probability event,* simultaneously for all $\pi \in \Pi$,

$$V(\pi) \le \hat{V}^{+,*}(\pi) + 2C_v \left( R_n(\Pi) + \tfrac{5}{2}\sqrt{\tfrac{1}{2n}\log\tfrac{2}{\delta}} \right). \tag{126}$$

*Proof.* We start by bounding the sharp upper bound $V^{+,*}(\pi)$ of the value function. By assumption, we have that $|Y| \le C_y$. Hence, for any $\pi \in \Pi$, we can bound $V^{+,*}(\pi)$ via

$$|V^{+,*}(\pi)| \tag{127}$$

$$= \left| \int_{\mathcal{X}} \sum_a Q^{+,*}(a, x)\pi(a \mid x)\,\mathrm{d}p(x) \right| \tag{128}$$

$$\le \left| \sup_{(x,a)\in\mathcal{X}\times\mathcal{A}} Q^{+,*}(a, x) \right| \tag{129}$$

$$= \sup_{(x,a)\in\mathcal{X}\times\mathcal{A}} \left| \left( (1 - \Gamma^{-1})e(a, x) + \Gamma^{-1} \right)\mathbb{E}\Big[Y\mathbb{1}_{\{Y \le F_{x,a}^{-1}(\alpha^+)\}} \mid X = x, A = a\Big] \right. \tag{130}$$

$$\left. + \left( (1 - \Gamma)e(a, x) + \Gamma \right)\mathbb{E}\Big[Y\mathbb{1}_{\{Y \ge F_{x,a}^{-1}(\alpha^+)\}} \mid X = x, A = a\Big] \right| \tag{131}$$

$$\le \sup_{(x,a)\in\mathcal{X}\times\mathcal{A}} C_y \left( \left( (1 - \Gamma^{-1})e(a, x) + \Gamma^{-1} \right) + \left( (1 - \Gamma)e(a, x) + \Gamma \right) \right) \tag{132}$$

$$\le C_y \left( 2 + 2\Gamma^{-1} + 2\Gamma \right) \tag{133}$$

$$= C_v. \tag{134}$$

Now, for the main result, we seek to find an upper bound for

$$\sup_{\pi \in \Pi} \hat{V}^{+,*}(\pi) - V(\pi). \tag{135}$$

By adding a zero and sublinearity of the supremum operator, we have that

$$\sup_{\pi \in \Pi} \hat{V}^{+,*}(\pi) - V(\pi) \le \sup_{\pi \in \Pi} \left( \hat{V}^{+,*}(\pi) - V^{+,*}(\pi) \right) + \sup_{\pi \in \Pi} \left( V^{+,*}(\pi) - V(\pi) \right). \tag{136}$$

Further, by validity of our bounds, we know that

$$\sup_{\pi \in \Pi} V^{+,*}(\pi) - V(\pi) \ge 0, \tag{137}$$

such that we only need to focus on

$$D = \sup_{\pi \in \Pi} \hat{V}^{+,*}(\pi) - V^{+,*}(\pi). \tag{138}$$

Since

$$\hat{V}^{+,*}(\pi) = \frac{1}{n}\sum_{i=1}^n V_i^{+,*}(\pi) = \frac{1}{n}\sum_{i=1}^n \sum_a Q^{+,*}(X_i, a)\pi(a \mid X_i) \tag{139}$$

is a sample average with $|V_i^{+,*}(\pi)| \le C_v$, we know that $D$ satisfies the bounded difference with $C_v/n$. Hence, we can apply McDiarmid's inequality (McDiarmid, 1989), which yields

$$\mathbb{P}\Big( D - \mathbb{E}[D] \ge \epsilon \Big) \le \underbrace{\exp\Big( -\frac{2\epsilon^2 n}{C_v^2} \Big)}_{=p_1}. \tag{140}$$

Then, solving for $\epsilon$ gives us

$$\epsilon = \sqrt{\frac{C_v^2}{2n} \log\left(\frac{1}{p_1}\right)}. \tag{141}$$

Hence, we know, with probability at least $1 - p_1$, that

$$D \leq \mathbb{E}[D] + \sqrt{\frac{C_v^2}{2n} \log\left(\frac{1}{p_1}\right)}. \tag{142}$$

As in (Athey & Wager, 2021; Frauen et al., 2024b; Hatt et al., 2022; Kallus & Zhou, 2018a), we express the flexibility of our policy class $\Pi$ in terms of the Rademacher complexity. For this, we first note that a standard symmetrization argument yields

$$\mathbb{E}[D] \leq \mathbb{E}\left[\frac{1}{2^n} \sum_{\sigma \in \{-1, +1\}} \sup_{\pi \in \Pi} \left| \frac{1}{n} \sum_{i=1}^{n} \sigma_i V_i^{+,*}(\pi) \right| \right], \tag{143}$$

where $\sigma_i \sim_{\text{iid}} \text{Unif}\{-1, +1\}$. Then, using the Rademacher-comparison lemma (Ledoux & Talagrand, 1989), we have that

$$\mathbb{E}[D] \leq 2C_v \mathbb{E}\left[\mathcal{R}_n(\Pi)\right], \tag{144}$$

where

$$\mathcal{R}_n(\Pi) = \mathbb{E}_\sigma\left[\sup_{\pi \in \Pi} \frac{1}{n} \sum_{i=1}^{n} \sigma_i V_i^{+,*}\right] \tag{145}$$

is the empirical Rademacher complexity of policy class $\Pi$. Again, $\mathcal{R}_n(\Pi)$ satisfies bounded difference with $2/n$, such that we can apply McDiarmid's inequality (McDiarmid, 1989). This gives

$$\mathbb{P}\left(\mathcal{R}_n(\Pi) - \mathbb{E}[\mathcal{R}_n(\Pi)] \geq \epsilon\right) \leq \underbrace{\exp\left(-\frac{\epsilon^2 n}{2}\right)}_{=p_2}. \tag{146}$$

Solving for $\epsilon$ yields

$$\epsilon = \sqrt{\frac{2}{n} \log\left(\frac{1}{p_2}\right)}, \tag{147}$$

such that, with probability at least $1 - p_2$, we have that

$$\mathbb{E}\left[\mathcal{R}_n(\Pi)\right] \leq \mathcal{R}_n(\Pi) + \sqrt{\frac{2}{n} \log\left(\frac{1}{p_2}\right)}. \tag{148}$$

Combining Equation 148 with our previous result in Equation 142, we have, with probability of at least $1 - p_1 - p_2$, that

$$\sup_{\pi \in \Pi} \left(\hat{V}^{+,*}(\pi) - V^{+,*}(\pi)\right) \tag{149}$$

$$\leq 2C_v \mathcal{R}_n(\Pi) + \sqrt{\frac{C_v^2}{2n} \log\left(\frac{1}{p_1}\right)} + 2C_v \sqrt{\frac{2}{n} \log\left(\frac{1}{p_2}\right)} \tag{150}$$

$$= 2C_v\left(\mathcal{R}_n(\Pi) + \sqrt{\frac{1}{8n} \log\left(\frac{1}{p_1}\right)} + \sqrt{\frac{2}{n} \log\left(\frac{1}{p_2}\right)}\right). \tag{151}$$

Now let $p_1 = p_2 = \delta/2$. Then, we know that with probability at least $1 - \delta$,

$$\sup_{\pi \in \Pi} \left(\hat{V}^{+,*}(\pi) - V^{+,*}(\pi)\right) \tag{152}$$

$$\leq 2C_v\left(\mathcal{R}_n(\Pi) + \sqrt{\frac{1}{8n} \log\left(\frac{1}{\delta}\right)} + \sqrt{\frac{2}{n} \log\left(\frac{1}{\delta}\right)}\right) \tag{153}$$

$$= 2C_v\left(\mathcal{R}_n(\Pi) + \frac{5}{2}\sqrt{\frac{1}{2n} \log\left(\frac{2}{\delta}\right)}\right), \tag{154}$$

or, equivalently,

$$V^{+,*}(\pi) \leq \hat{V}^{+,*}(\pi) + 2C_v \left( \mathcal{R}_n(\Pi) + \frac{5}{2} \sqrt{\frac{1}{2n} \log \left( \frac{2}{\delta} \right)} \right) \tag{155}$$

for all $\pi \in \Pi$, which concludes the proof. □

### F.4 BOUND OF THE REGRET FUNCTION

**Corollary F.4.** *Given $Q^{\pm,*}(a,x)$ and our sensitivity constraints $\mathcal{P}(\Gamma)$ as in Proposition 4.1, an upper bound for the regret function $R_{\pi_0}(\pi)$ is given by*

$$R_\pi^+(\pi) = \int_{\mathcal{X}} \sum_a \Big( Q^{+,*}(a,x)\pi(a\mid x) - Q^{-,*}(a,x)\pi_0(a\mid x) \Big)\, \mathrm{d}p(x). \tag{156}$$

*Proof.* Our proof follows similar steps as in the proof of Proposition 4.1. For clarity, we repeat the same steps such that the proof is self-contained.

Again, we start by noting that the upper bound on the regret function depends on the choice of our sensitivity constraints and, hence, the set of distributions $\mathcal{P}(\Gamma)$. Therefore, we can write that

$$R_{\pi_0}^+(\pi) = R_{\pi_0}^+(\pi, \mathcal{P}(\Gamma)). \tag{157}$$

Following similar steps as in Proposition 4.1, we can write

$$R_{\pi_0}^+(\pi) \tag{158}$$

$$= R_{\pi_0}^+(\pi, \mathcal{P}(\Gamma)) \tag{159}$$

$$= \sup_{\tilde{p}\in\mathcal{P}(\Gamma)} R_{\pi_0}(\pi, \tilde{p}) \tag{160}$$

$$= \sup_{\tilde{p}\in\mathcal{P}(\Gamma)} \Big( V(\pi, \tilde{p}) - V(\pi_0, \tilde{p}) \Big) \tag{161}$$

$$= \sup_{\tilde{p}\in\mathcal{P}(\Gamma)} \int_{\mathcal{X}} \sum_a \Big( Q(a,x,\tilde{p})\pi(a\mid x) - Q(a,x,\tilde{p})\pi_0(a\mid x) \Big)\, \mathrm{d}\tilde{p}(x) \tag{162}$$

$$= \sup_{\tilde{p}\in\mathcal{P}(\Gamma)} \int_{\mathcal{X}} \sum_a \Big( Q(a,x,\tilde{p})\pi(a\mid x) - Q(a,x,\tilde{p})\pi_0(a\mid x) \Big)\, \mathrm{d}p(x), \tag{163}$$

where Equation 163 again follows from $p(\mathcal{D}) = \tilde{p}(\mathcal{D})$ for all $\tilde{p}\in\mathcal{P}(\Gamma)$.

Since the optimal bounds $Q^{\pm,*}(a,x)$ are those $Q(a,x,\tilde{p})$, $\tilde{p}\in\mathcal{P}(\Gamma)$, for which the supremum/infimum are attained, we have that

$$Q(a,x,\tilde{p}) \le \sup_{\tilde{p}\in\mathcal{P}(\Gamma)} Q(a,x,\tilde{p}) = Q^{+,*}(a,x,\mathcal{P}(\Gamma)) \tag{164}$$

and

$$Q(a,x,\tilde{p}) \ge \inf_{p\in\mathcal{P}(\Gamma)} Q(a,x,\tilde{p}) = Q^{-,*}(a,x,\mathcal{P}(\Gamma)) \tag{165}$$

for all $\tilde{p}\in\mathcal{P}(\Gamma)$. Then, since $Q^{\pm,*}(a,x)\in L^1(\pi,p)$, it follows by dominated convergence that

$$\sup_{\tilde{p}\in\mathcal{P}(\Gamma)} \int_{\mathcal{X}} \sum_a \Big( Q(a,x,\tilde{p})\pi(a\mid x) - Q(a,x,\tilde{p})\pi_0(a\mid x) \Big)\, \mathrm{d}p(x) \tag{166}$$

$$= \int_{\mathcal{X}} \sup_{\tilde{p}\in\mathcal{P}(\Gamma)} \sum_a \Big( Q(a,x,\tilde{p})\pi(a\mid x) - Q(a,x,\tilde{p})\pi_0(a\mid x) \Big)\, \mathrm{d}p(x) \tag{167}$$

$$\le \int_{\mathcal{X}} \sum_a \sup_{\tilde{p}\in\mathcal{P}(\Gamma)} Q(a,x,\tilde{p})\pi(a\mid x)\, \mathrm{d}p(x) - \int_{\mathcal{X}} \sum_a \inf_{\tilde{p}\in\mathcal{P}(\Gamma)} Q(a,x,\tilde{p})\pi_0(a\mid x)\, \mathrm{d}p(x) \tag{168}$$

$$= \int_{\mathcal{X}} \sum_a \Big( Q^{+,*}(a,x)\pi(a\mid x) - Q^{-,*}(a,x)\pi_0(a\mid x) \Big)\, \mathrm{d}p(x), \tag{169}$$

where Equation 168 follows from the sublinearity of the supremum/infimum operator. $\square$

### F.5 EFFICIENT ESTIMATOR OF THE REGRET BOUND

**Corollary F.5.** *Let $\mathbb{E}[|Y|^2] < \infty$ and $p(y \mid x, a)$ admit a continuous density bounded away from zero in a neighborhood of $F_{x,a}^{-1}(\alpha^+)$. This holds, for example, if the density $p(y \mid x, a)$ is strictly positive and continuous. Then, an efficient estimator for the upper bound of the regret function is given by*

$$\hat{R}_{\pi_0}^+(\pi) \tag{170}$$

$$= \sum_{\pm \in \{-,+\}} \pm \mathbb{P}_n \Big\{ \sum_a \pi_{a,X}^{\pm} \Big[ \hat{Q}_{a,X}^{\pm,*} - \hat{e}_{a,X} \Big( b^{\mp} \underline{\hat{\mu}}_{a,X}^{\pm} + b^{\pm} \hat{\bar{\mu}}_{a,X}^{\pm} \Big) \Big] + \pi_{A,X}^{\pm} \Big( b^{\mp} \underline{\hat{\mu}}_{A,X}^{\pm} + b^{\pm} \hat{\bar{\mu}}_{A,X}^{\pm} \Big) \tag{171}$$

$$+ \frac{\pi_{A,X}^{\pm}}{\hat{e}_{A,X}} \Big[ \Big( \hat{\underline{c}}_{A,X}^{\mp} - \hat{\bar{c}}_{A,X}^{\pm} \Big) \Big( \hat{F}_{X,A}^{-1}(\alpha^{\pm})(\underline{\hat{\Delta}}_{Y,A,X}^{\pm} - \alpha^{\pm}) \Big) + \hat{\underline{c}}_{A,X}^{\mp} \Big( Y \underline{\hat{\Delta}}_{Y,A,X}^{\pm} - \underline{\hat{\mu}}_{A,X}^{\pm} \Big) + \hat{\bar{c}}_{A,X}^{\pm} \Big( Y \hat{\bar{\Delta}}_{Y,A,X}^{\pm} - \hat{\bar{\mu}}_{A,X}^{\pm} \Big) \Big] \Big\}, \tag{172}$$

*where we use $\pi^+ = \pi$ and $\pi^- = \pi_0$ for readability.*

*Proof.* The upper bound of the regret function is given by

$$R_{\pi_0}^+ = V^{+,*}(\pi) - V^{-,*}(\pi_0), \tag{173}$$

where

$$V^{\pm,*}(\pi) = \int_{\mathcal{X}} \sum_a Q^{\pm,*}(a, x) \pi(a \mid x) \, \mathrm{d}p(x). \tag{174}$$

By additivity of the efficient influence function, we know that

$$\mathbb{IF}\Big(R_{\pi_0}^{+,*}(\pi)\Big) = \mathbb{IF}\Big(V^{+,*}(\pi)\Big) - \mathbb{IF}\Big(V^{-,*}(\pi_0)\Big), \tag{175}$$

such that we can focus on both terms separately in Supplements F.5.1 and F.5.2 and then plug them together in Supplement F.5.3 in order to obtain our efficient estimator.

### F.5.1 EFFICIENT INFLUENCE FUNCTION OF $V^{+,*}(\pi)$

We already have the efficient influence function of $V^{+,*}(\pi)$ from the proof of Theorem 4.3 in Supplement F.2. It is given by

$$\mathbb{IF}\Big(V^{+,*}(\pi; \eta)\Big) \tag{176}$$

$$= -V^{+,*}(\pi) + \sum_a \pi(a \mid X) \Big[ Q^{+,*}(a, X) - e(a, X) \Big( b^- \underline{\mu}^+(a, X; \eta) + b^+ \bar{\mu}^+(a, X; \eta) \Big) \Big] \tag{177}$$

$$+ \pi(A \mid X) \Big( b^- \underline{\mu}^+(A, X; \eta) + b^+ \bar{\mu}^+(A, X; \eta) \Big) \tag{178}$$

$$+ \frac{\pi(A \mid X)}{e(A, X)} \Big[ \Big( c^-(A, X; \eta) - c^+(A, X; \eta) \Big) \Big( F_{X,A}^{-1}(\alpha^+)(\underline{\Delta}_{\alpha^+}(Y, A, X; \eta) - \alpha^+) \Big) \tag{179}$$

$$+ c^-(A, X; \eta) \Big( Y \underline{\Delta}^+(Y, A, X; \eta) - \underline{\mu}^+(A, X; \eta) \Big) + c^+(A, X; \eta) \Big( Y \bar{\Delta}^+(Y, A, X; \eta) - \bar{\mu}^+(A, X; \eta) \Big) \Big]. \tag{180}$$

### F.5.2 EFFICIENT INFLUENCE FUNCTION OF $V^{-,*}(\pi)$

We can derive the sharp lower bound for the value function analogously to $V^{+,*}$. Again, we make use of the chain rule for deriving efficient influence function (Kennedy, 2022), a proof of which can be found in (Luedtke, 2024) **(Lemma S3)**.

For this, let $\alpha^- = 1/(1 + \Gamma)$. Then, the sharp lower bound of the CAPO is given by

$$Q^{-,*}(a, x; \eta) \tag{181}$$

$$= \Big( (1 - \Gamma)e(a, x) + \Gamma \Big) \mathbb{E}\Big[ Y \mathbb{1}_{\{Y \leq F_{X,A}^{-1}(\alpha^-)\}} \mid X = x, A = a \Big] \tag{182}$$

$$+ \Big( (1 - \Gamma^{-1})e(a, x) + \Gamma^{-1} \Big) \mathbb{E}\Big[ Y \mathbb{1}_{\{Y \geq F_{X,A}^{-1}(\alpha^-)\}} \mid X = x, A = a \Big]. \tag{183}$$

Hence, the influence function of $Q^{-,*}(a, x)$ is given by

$$\mathbb{IF}\Big( Q^{-,*}(a, x; \eta) \Big) \tag{184}$$

$$= \frac{\mathbb{1}_{\{X=x\}}}{p(x)} (1 - \Gamma)\Big( \mathbb{1}_{\{A=a\}} - e(a, x) \Big) \mathbb{E}\Big[ Y \mathbb{1}_{\{Y \leq F_{X,A}^{-1}(\alpha^-)\}} \mid X = x, A = a \Big] \tag{185}$$

$$+ \frac{\mathbb{1}_{\{X=x,A=a\}}}{p(a, x)} \Big( (1 - \Gamma)e(a, x) + \Gamma \Big) \tag{186}$$

$$\times \Big( Y \mathbb{1}_{\{Y \leq F_{x,a}^{-1}(\alpha^-)\}} - \mathbb{E}[Y \mathbb{1}_{\{Y \leq F_{X,A}^{-1}(\alpha^-)\}} \mid X = x, A = a] + F_{x,a}^{-1}(\alpha^-)(\mathbb{1}_{\{Y \leq F_{x,a}^{-1}(\alpha^-)\}} - \alpha^-) \Big) \tag{187}$$

$$+ \frac{\mathbb{1}_{\{X=x\}}}{p(x)} (1 - \Gamma^{-1})\Big( \mathbb{1}_{\{A=a\}} - e(a, x) \Big) \mathbb{E}\Big[ Y \mathbb{1}_{\{Y \geq F_{X,A}^{-1}(\alpha^-)\}} \mid X = x, A = a \Big] \tag{188}$$

$$+ \frac{\mathbb{1}_{\{X=x,A=a\}}}{p(a, x)} \Big( (1 - \Gamma^{-1})e(a, x) + \Gamma^{-1} \Big) \tag{189}$$

$$\times \Big( Y \mathbb{1}_{\{Y \geq F_{x,a}^{-1}(\alpha^-)\}} - \mathbb{E}[Y \mathbb{1}_{\{Y \geq F_{X,A}^{-1}(\alpha^-)\}} \mid X = x, A = a] - F_{x,a}^{-1}(\alpha^-)(\mathbb{1}_{\{Y \leq F_{x,a}^{-1}(\alpha^-)\}} - \alpha^-) \Big) \tag{190}$$

$$= \frac{\mathbb{1}_{\{X=x\}}}{p(x)} \Big\{ \Big( \mathbb{1}_{\{A=a\}} - e(a, x) \Big) \Big( b^+ \underline{\mu}^-(a, x; \eta) + b^- \bar{\mu}^-(a, x; \eta) \Big) \tag{191}$$

$$+ \frac{\mathbb{1}_{\{A=a\}}}{e(a, x)} \Big[ \Big( c^+(a, x; \eta) - c^-(a, x; \eta) \Big) \Big( F_{x,a}^{-1}(\alpha^-)(\underline{\Delta}^-(Y, a, x; \eta) - \alpha^-) \Big) \tag{192}$$

$$+ c^+(a, x; \eta)\Big( Y \underline{\Delta}^-(Y, a, x; \eta) - \underline{\mu}^-(a, x; \eta) \Big) + c^-(a, x; \eta)\Big( Y \bar{\Delta}^-(Y, a, x; \eta) - \bar{\mu}^-(a, x; \eta) \Big) \Big] \Big\} \tag{193}$$

Following the same steps as for Equation 45, the influence function of $V^{-,*}(\pi)$ is given by

$$\mathbb{IF}\Big( V^{-,*}(\pi; \eta) \Big) \tag{194}$$

$$= -V^{-,*}(\pi; \eta) + \sum_a \pi(a \mid X)\Big[ Q^{-,*}(a, X; \eta) - e(a, X)\Big( b^+ \underline{\mu}^-(a, X; \eta) + b^- \bar{\mu}^-(a, X; \eta) \Big) \Big] \tag{195}$$

$$+ \pi(A \mid X)\Big( b^+ \underline{\mu}^-(A, X; \eta) + b^- \bar{\mu}^-(A, X; \eta) \Big) \tag{196}$$

$$+ \frac{\pi(A \mid X)}{e(A, X)} \Big[ \Big( c^+(A, X; \eta) - c^-(A, X; \eta) \Big)\Big( F_{X,A}^{-1}(\alpha^-)(\underline{\Delta}^-(Y, A, X; \eta) - \alpha^-) \Big) \tag{197}$$

$$+ c^+(A, X; \eta)\Big( Y \underline{\Delta}^-(Y, A, X; \eta) - \underline{\mu}^-(A, X; \eta) \Big) + c^-(A, X; \eta)\Big( Y \bar{\Delta}^-(Y, A, X; \eta) - \bar{\mu}^-(A, X; \eta) \Big) \Big]. \tag{198}$$

### F.5.3 EFFICIENT ESTIMATOR OF $R_{\pi_0}^+(\pi)$

We can derive the efficient estimator for the bounds of the value function through one-step bias correction using our results form Supplements F.5.1 and F.5.2 via

$$V^{\pm,*}(\pi;\hat{\eta}) + \mathbb{P}_n\Big\{V^{\pm,*}(\pi;\hat{\eta})\Big\} \tag{199}$$

$$=\mathbb{P}_n\Big\{\sum_a \pi(a\mid X)\Big[Q^{\pm,*}(a,X;\hat{\eta}) - \hat{e}(a,X)\Big(b^{\mp}\underline{\hat{\mu}}^{\pm}(a,X;\hat{\eta}) + b^{\pm}\hat{\bar{\mu}}^{\pm}(a,X;\hat{\eta})\Big)\Big] \tag{200}$$

$$+ \pi(A\mid X)\Big(b^{\mp}\underline{\hat{\mu}}^{\pm}(A,X;\hat{\eta}) + b^{\pm}\hat{\bar{\mu}}^{\pm}(A,X;\hat{\eta})\Big) \tag{201}$$

$$+ \frac{\pi(A\mid X)}{\hat{e}(A,X)}\Big[\Big(c^{\mp}(A,X;\hat{\eta}) - c^{\pm}(A,X;\hat{\eta})\Big)\Big(\hat{F}_{X,A}^{-1}(\alpha^{\pm})(\underline{\Delta}^{\pm}(Y,A,X;\hat{\eta}) - \alpha^{\pm})\Big) \tag{202}$$

$$+ c^{\mp}(A,X;\hat{\eta})\Big(Y\underline{\Delta}^{\pm}(Y,A,X;\hat{\eta}) - \underline{\hat{\mu}}^{\pm}(A,X;\hat{\eta})\Big) + c^{\pm}(A,X;\hat{\eta})\Big(Y\bar{\Delta}^{\pm}(Y,A,X;\hat{\eta}) - \hat{\bar{\mu}}^{\pm}(A,X;\hat{\eta})\Big)\Big]\Big\} \tag{203}$$

$$=\mathbb{P}_n\Big\{\sum_a \pi_{a,X}\Big[\hat{Q}_{a,X}^{\pm,*} - \hat{e}_{a,X}\Big(b^{\mp}\underline{\hat{\mu}}_{a,X}^{\pm} + b^{\pm}\hat{\bar{\mu}}_{a,X}^{\pm}\Big)\Big] + \pi_{A,X}\Big(b^{\mp}\underline{\hat{\mu}}_{A,X}^{\pm} + b^{\pm}\hat{\bar{\mu}}_{A,X}^{\pm}\Big) \tag{204}$$

$$+ \frac{\pi_{A,X}}{\hat{e}_{A,X}}\Big[\Big(\hat{c}_{A,X}^{\mp} - \hat{c}_{A,X}^{\pm}\Big)\Big(\hat{F}_{X,A}^{-1}(\alpha^{\pm})(\underline{\hat{\Delta}}_{Y,A,X}^{\pm} - \alpha^{\pm})\Big) + \hat{c}_{A,X}^{\mp}\Big(Y\underline{\hat{\Delta}}_{Y,A,X}^{\pm} - \underline{\hat{\mu}}_{A,X}^{\pm}\Big) + \hat{c}_{A,X}^{\pm}\Big(Y\hat{\bar{\Delta}}_{Y,A,X}^{\pm} - \hat{\bar{\mu}}_{A,X}^{\pm}\Big)\Big]\Big\} \tag{205}$$

using our short-hand notation from the main paper. Hence, the efficient estimator of the upper bound of the regret function is given by

$$\hat{R}_{\pi_0}^+(\pi) \tag{206}$$

$$=R_{\pi_0}^+(\pi;\hat{\eta}) + \mathbb{P}_n\Big\{\mathbb{IF}\Big(R_{\pi_0}^{+,*}(\pi;\hat{\eta})\Big)\Big\} \tag{207}$$

$$=\Big(V^{+,*}(\pi;\hat{\eta}) - V^{-,*}(\pi_0;\hat{\eta})\Big) + \mathbb{P}_n\Big\{\mathbb{IF}\Big(V^{+,*}(\pi;\hat{\eta})\Big) - \mathbb{IF}\Big(V^{-,*}(\pi_0;\hat{\eta})\Big)\Big\} \tag{208}$$

$$= \sum_{\pm\in\{+,-\}} \pm\mathbb{P}_n\Big\{\sum_a \pi_{a,X}^{\pm}\Big[\hat{Q}_{a,X}^{\pm,*} - \hat{e}_{a,X}\Big(b^{\mp}\underline{\hat{\mu}}_{a,X}^{\pm} + b^{\pm}\hat{\bar{\mu}}_{a,X}^{\pm}\Big)\Big] + \pi_{A,X}^{\pm}\Big(b^{\mp}\underline{\hat{\mu}}_{A,X}^{\pm} + b^{\pm}\hat{\bar{\mu}}_{A,X}^{\pm}\Big) \tag{209}$$

$$+ \frac{\pi_{A,X}^{\pm}}{\hat{e}_{A,X}}\Big[\Big(\hat{c}_{A,X}^{\mp} - \hat{c}_{A,X}^{\pm}\Big)\Big(\hat{F}_{X,A}^{-1}(\alpha^{\pm})(\underline{\hat{\Delta}}_{Y,A,X}^{\pm} - \alpha^{\pm})\Big) + \hat{c}_{A,X}^{\mp}\Big(Y\underline{\hat{\Delta}}_{Y,A,X}^{\pm} - \underline{\hat{\mu}}_{A,X}^{\pm}\Big) + \hat{c}_{A,X}^{\pm}\Big(Y\hat{\bar{\Delta}}_{Y,A,X}^{\pm} - \hat{\bar{\mu}}_{A,X}^{\pm}\Big)\Big]\Big\}, \tag{210}$$

where we let $\pi^+ = \pi$ and $\pi^- = \pi_0$ for readability. $\qquad\square$

### F.6 Improvement Guarantee: Regret Function

**Corollary F.6.** *Under the same assumption as in Theorem 4.4, simultaneously for all $\pi \in \Pi$ and fixed baseline policy $\pi_0 \in \Pi$, it holds, with probability $1 - \delta$, that*

$$R_{\pi_0}(\pi) \leq \hat{R}_{\pi_0}^{+}(\pi) + 4C_v\left(\mathcal{R}_n(\Pi) + \frac{5}{2}\sqrt{\frac{1}{2n}\log\left(\frac{2}{\delta}\right)}\right), \tag{211}$$

*where $C_v = 2C_y(1 + \Gamma^{-1} + \Gamma)$ and $\mathcal{R}_n(\pi)$ is the empirical Rademacher complexity of policy class $\Pi$.*

*Proof.* In order to show the main result, we note that

$$R_{\pi_0}^{+}(\pi) = V^{+,*}(\pi) - V^{-,*}(\pi_0) \tag{212}$$

for arbitrary $\pi, \pi_0 \in \Pi$.

From Theorem 4.4, we know that

$$V^{+,*}(\pi) \leq \hat{V}^{+,*}(\pi) + 2C_v\left(\mathcal{R}_n(\Pi) + \frac{5}{2}\sqrt{\frac{1}{2n}\log\left(\frac{2}{\delta}\right)}\right). \tag{213}$$

Since $\pi, \pi_0 \in \Pi$ are arbitrary, we can repeat the same arguments for $V^{-,*}(\pi_0)$ and obtain

$$V^{-,*}(\pi_0) \geq \hat{V}^{-,*}(\pi_0) - 2C_v\left(\mathcal{R}_n(\Pi) + \frac{5}{2}\sqrt{\frac{1}{2n}\log\left(\frac{2}{\delta}\right)}\right). \tag{214}$$

Then, we conclude the proof by

$$R_{\pi_0}(\pi) \tag{215}$$
$$= V(\pi) - V(\pi_0) \tag{216}$$
$$\leq V^{+,*}(\pi) - V^{-,*}(\pi_0) \tag{217}$$
$$= \left[\hat{V}^{+,*}(\pi) + 2C_v\left(\mathcal{R}_n(\Pi) + \frac{5}{2}\sqrt{\frac{1}{2n}\log\left(\frac{2}{\delta}\right)}\right)\right] - \left[\hat{V}^{-,*}(\pi_0) - 2C_v\left(\mathcal{R}_n(\Pi) + \frac{5}{2}\sqrt{\frac{1}{2n}\log\left(\frac{2}{\delta}\right)}\right)\right] \tag{218}$$
$$= \hat{R}_{\pi_0}^{+}(\pi) + 4C_v\left(\mathcal{R}_n(\Pi) + \frac{5}{2}\sqrt{\frac{1}{2n}\log\left(\frac{2}{\delta}\right)}\right). \tag{219}$$

$\square$

## G   IMPLEMENTATION DETAILS

We summarize the neural instantiations of all estimators in Section 5.

**Runtime:** Training the first and second stage models for our method took in total approximately 20 seconds using $n = 1000$ synthetic data samples and a standard computer with AMD Ryzen 7 Pro CPU and 32GB of RAM.634. All baselines have a comparable runtime.

| Nuisance function | Hyperparameter | Configuration | Standard methods | | Kallus & Zhou (2018a; 2021) | Plug-in sharp (ours) | Efficient sharp (ours) |
|---|---|---|---|---|---|---|---|
| | | | IPW estimator | DR estimator | | | |
| Propensity score | Hidden layers
Layer size
Hidden activation
Learning rate
Number of epochs
Early stopping patience
Batch size | 3
$\{64, 64, 32\}$
ReLU
0.001
300
10
64 | ✓ | ✓ | ✓ | ✗ | ✓ |
| Conditional quantile function | Hidden layers
Layer size
Hidden activation
Learning rate
Number of epochs
Early stopping patience
Batch size | 3
$\{64, 64, 32\}$
ReLU
0.001
300
10
64 | ✗ | ✗ | ✗ | ✗ | ✓ |
| (Truncated) outcome regression model | Hidden layers
Layer size
Hidden activation
Learning rate
Number of epochs
Early stopping patience
Batch size | 3
$\{64, 64, 32\}$
ReLU
0.001
300
10
64 | ✗ | ✓ | ✗ | ✓ | ✓ |
| Parametric policy | Hidden layers
Layer size
Hidden activation
Learning rate
Number of epochs
Early stopping patience
Batch size | 3
$\{64, 64, 32\}$
ReLU
0.001
300
10
64 | ✓ | ✓ | ✓ | ✓ | ✓ |

Table 4: Neural instantiations of estimated nuisance functions $\hat{\eta}$ and parametric policy $\pi_\theta$. To ensure a fair comparison, all methods share the same nuisance function where applicable. For all models, we set the split parameter for training the nuisance and the policy model to $\rho = 0.5$.

