# OpenReview forum: "Efficient and Sharp Off-Policy Learning under Unobserved Confounding"
_ICLR.cc/2026/Conference — ICLR 2026 Poster_

### Official Review · Reviewer_P2Zy · 2025-10-30

**Soundness:** 3
**Presentation:** 3
**Contribution:** 3
**Rating:** 6
**Confidence:** 3

**Summary:**

The paper derives closed-form sharp bounds for policy value under MSM, plus a semi-parametrically efficient estimator. It avoids unstable minimax/IPW and proves optimal confounding-robust policy learning.

**Strengths:**

- Closed-form sharp bound for value under MSM
- One-step bias-corrected estimator hits the efficiency bound
- Learning guarantees to the optimal confounding-robust policy

**Weaknesses:**

- The EIF and one-step estimator rely on quantiles $F_{x,a}^{-1}(\alpha_{\pm})$. You do not state standard conditions ensuring pathwise differentiability.
- You claims the estimator “is semi-parametrically efficient” and points to D.2, which provides an influence function expression and cites a chain-rule lemma. But you never identify the canonical gradient in the nonparametric model nor verify your influence function equals it.
- Theorem 4.4 needs a uniform bound, but the current version is pointwise in $\pi$.
- The nuisance $\eta$ in (14) includes the quantiles, but your EIF contains terms like $(\Delta-\alpha)F^{-1}$ that rely on differentiability of these nuisances. Please clarify.
- Before Theorem 4.4, you write “parametric policy classes (e.g., neural networks) have vanishing $R_n(\Pi)\in O(n^{-1/2})$". For neural networks, this needs norm constraints, otherwise $R_n$ need not decay at root-n rate.

**Questions:**

- In Algorithm 1, Step 6 says Estimate $V^{+,*}$ as in (2), but (2) just defines the propensity.

---

> ### Author Response · Authors · 2025-11-17
>
> Thank you very much for your positive review of our paper!
>
>
> ### Responses to Weaknesses:
>
> **> W1: Standard conditions for pathwise differentiability**
>
> Thank you for pointing this out! You are correct in that regarding the quantile functions and the truncated moments, there are indeed non-trivial assumptions to be made. We apologize for this oversight.
>
> **Conditional quantiles:**
>
> For our derivations of the efficient influence function, we require in Eq (68) to Eq (78) that (1) the conditional quantile function is **unique** (Eq (73)), and (2) that the **conditional density function** of the outcome is **strictly positive** (Eq (74) to Eq (77)). This aligns with the assumptions made by Firpo (2007).
>
> To be even more precise, for our proof, we require that $p(y|x,a)$ admits continuous density strictly bounded away from zero in a neighbourhood of $F_{x,a}^{−1}(\alpha^+)$.
>
> **Action:** We add the following assumption to our **revised Theorem 4.3** in order to ensure pathwise differentiability as follows in our paper:
>
> “Let $p(y\mid x,a)$ admit a continuous density strictly bounded away from zero in a neighborhood of $F_{x,a}^{-1}(\alpha^+)$. This holds, for example, if the conditional density $p(y\mid x,a)$ is strictly positive and continuous.”
>
> We now further **highlight** this crucial assumption in our **revised proof** of Theorem 4.3.
>
> **Conditional expected truncated moments:**
>
> We further need to guarantee that the conditional expected truncated moments (the $\Delta$ terms) are pathwise differentiable.
>
> For these terms to be well defined throughout the derivation of the CAPO EIF (more specifically, in order to apply the chain rule in the expression in Eq (53)), we need that $Y$ admits a finite first moment.
>
> Further, for our one-step estimator to be **semi-parametrically efficient**, we need a central limit theorem to apply and, hence, require $Y$ to be square integrable / to have a **finite second moment**.
>
> **Action:** We further add a finite first moment assumption of $Y$ to our assumption in our **revised Theorem 4.3** to guarantee that the estimator exists. We add a second square-integrability assumption to guarantee that the estimator achieves semi-parametric efficiency. We further highlight where we use the first condition in our **revised proofs**.
>
> >**W2: Canonical gradient**
>
> Thank you. You are correct that semiparametric efficiency requires showing that the influence function we derive is in fact the *canonical gradient* of the parameter in the nonparametric model.
>
> To clarify: The canonical gradient is defined as the unique element of the tangent space $\mathcal{T}(P)$ that represents the pathwise derivative of the parameter mapping
> $$
> \Psi(P)=V^{+,\*}(\pi;P),$$
> i.e.
> $\frac{d}{d \epsilon}\Psi(P_\epsilon) |_{\epsilon=0}=$
>
> $\mathbb{E}[g^ {\*}(Z) s_\epsilon(Z)]
> $
> for all scores $s_\epsilon \in \mathcal{T}(P)$
>
> and the canonical gradient $g^\*$ is the **Riesz representer** of this linear functional (see, e.g., van der Vaart, 1998).
>
> In our proofs, we compute the pathwise derivative explicitly using the chain-rule construction for pathwise differentiability (Kennedy, 2022; Lemma S3 in Luedtke, 2024).
>
> Here, we decompose $V^{+,*}(\pi)$ into elementary building blocks such as conditional expectations and conditional distributions, each of which has a known canonical gradient. For every building block, we then insert its known efficient influence function, apply the chain rule, and simplify the resulting expression.
>
> This procedure is, to our knowledge, the standard way to recover the canonical gradient in a nonparametric model (e.g., Melnychuk et al., 2024): the output of the chain–rule construction is exactly the unique function $\varphi(Z)$ satisfying
> $$
> \frac{d}{d\epsilon}\Psi(P_\epsilon)|_{\epsilon=0}=$$
>
> $$
> \mathbb{E} [\varphi(Z) s_\varepsilon(Z)]
> $$
> for every regular submodel $\{P_\epsilon\}$.
>
> The *output of the chain-rule construction is – by definition – the canonical gradient of the composite parameter*.  In other words, the influence function we present is the unique element in the tangent space that represents the pathwise derivative of $V^{+,*}(\pi)$, and hence equals the canonical gradient.
>
> By standard semiparametric theory (Kennedy, 2022; van der Vaart, 1998), the one–step estimator based on this $\varphi$ is therefore semiparametrically efficient for $V^{+,*}(\pi)$.
>
> **Action:** We add a foreword on the construction of semi-parametric estimators in our **revised Supplement G** to make the above identification explicit, and we cite the relevant results (Kennedy, 2022; Luedtke, 2024; van der Vaart, 1998) which show that the chain-rule construction yields the canonical gradient (i.e., the efficient influence function) for composite pathwise differentiable functionals.

---

> ### Author Response · Authors · 2025-11-17
>
> **> W3: Theorem 4.4**
>
> Thank you for pointing this out! We agree the statement should be uniform; this was only an imprecise wording on our behalf. Our **proof already controls the supremum** over $\Pi$ via McDiarmid + symmetrization + Rademacher complexity; we now make this uniformity explicit in the theorem statement.
>
> **Action:** We **revised Theorem 4.4** to state a uniform high-probability bound that holds **simultaneously for all** $\pi\in\Pi$. The proof is unchanged in substance.
>
> **> W4: Differentiability of nuisances**
>
> Thank you! You are correct in that the conditional quantile function and the conditional expectation require additional assumptions. We kindly refer to our answer W1.
>
> **> W5: Parametric policy classes**
>
> Thank you for highlighting this!
>
> **Action:** We changed the sentence in our paper, where we now emphasize that neural networks are only root-n under the following condition:
>
> “For example, the class of neural networks with bounded size and smoothness, e.g., with fixed depth, bounded width, and weights controlled by simple norm constraints (so the networks are Lipschitz and outputs bounded), has Rademacher complexity $\mathcal{R}_n(\Pi)\in\mathcal{O}(n^{-1/2})$.”
>
>
> ### Response to Question:
>
> Thank you. This was a referencing error, which we fixed. It now correctly points to the equation in **Theorem 4.3.**
>
> ____
>
> Sergio Firpo. “Efficient Semiparametric Estimation of Quantile Treatment Effects.” *Econometrica* 75, no. 1 (2007): 259–76.
>
> Edward H. Kennedy. Semiparametric doubly robust targeted double machine learning: A review. arXiv preprint, 2022.
>
> Alex Luedtke. Simplifying debiased inference via automatic differentiation and probabilistic programming. arXiv preprint, 2405.08675, 2024.
>
> Valentyn Melnychuk, Stefan Feuerriegel, Mihaela van der Schaar. Quantifying Aleatoric Uncertainty of the Treatment Effect: A Novel Orthogonal Learner. In NeurIPS, 2024.
>
> Aart van der Vaart. Asymptotic statistics. Cambridge University Press, Cambridge, 1998. ISBN 0521496039.

---

> ### Author Response · Authors · 2025-11-28
>
> Dear reviewer P2Zy,
>
> We thank you for your helpful and positive feedback on our submission.
>
> We hope that our rebuttal could clarify your remaining question. Otherwise, we are happy to provide further insights!
>
> If you feel comfortable with it, we would highly appreciate it if you would consider increasing your score.

---

### Official Review · Reviewer_d6rY · 2025-10-31

**Soundness:** 3
**Presentation:** 3
**Contribution:** 2
**Rating:** 4
**Confidence:** 3

**Summary:**

This work offers a method for off-policy learning under unmeasured confounding, whereby unmeasured covariates can jointly affect treatment decisions and the outcome. Concretely, the authors propose a one-step bias-corrected estimator that estimates “sharp” (i.e., tightest possible) bounds around the true policy value under the marginal sensitivity model. These estimated bounds are then used for downstream policy learning. The authors show that their bias-corrected estimation approach is efficient — i.e., obtains the lowest variance among unbiased estimates — and relies upon a simplified minimization objective as compared to the mini-max style objective studied in prior work. The authors validate the approach theoretically and via experiments on synthetic and real-world data.

**Strengths:**

Policy learning under unmeasured confounding is an important problem with broad applications. The authors identify a key gap in the literature and address it via appropriate methods. Presentation of the work is effective: I especially appreciate Figure 2 illustrating how the concept of sharpness connects to regret. The theoretical results - including identification bounds (4.1), bias-corrected estimator (4.2), and learning guarantees (4.3) are also well suited to this problem setting. The synthetic + real-world empirical validation is also well-suited to the goals of the work.

**Weaknesses:**

## Connection to prior work & significance

In general, the authors provide solid coverage of prior work and appropriately situate the contribution in the literature. However, this work can be viewed as a targeted improvement on top of the basic framework established in Kallus & Zhou (2018a; 2021). While I still believe such work is valuable and worthy of publication, this somewhat limits the significance of the results.

More specifically, it would be helpful if the authors could provide more detailed technical discussion of the differences with (Kallus & Zhou, 2018a; 2021). How does instability in IPW weights propagate up to the estimated policy value/regret, and how is this solved by the proposed approach? Similarly, Kallus & Zhou (2018a; 2021) also show that the regret interval obtained under their proposed approach is sharp. While lines 211-226 offer helpful initial discussion, adding additional technical clarity would strengthen the discussion for the reader.

Further, Rambachan, Coston, and Kennedy (2022) derive sharp bounds for the policy value under a related Mean Outcome Sensitivity Model (MOSM), which they then estimate via a doubly-robust method. The authors further show that bounds on the MSOM imply MSM bounds. It would be helpful to outline similarities and differences with this approach. To start, I think the method proposed in this work generalizes to non-binary actions, and the bounds in  Rambachan, Coston, and Kennedy (2022) may not remain sharp w.r.t. the MSM after converting from the MOSM framework used in this work.

[1] Robust Design and Evaluation of Predictive Algorithms under Unobserved Confounding, https://arxiv.org/abs/2212.09844, Ashesh Rambachan, Amanda Coston, Edward Kennedy


## Empirical validation


The general empirical validation of the work is sound, and the authors demonstrate that the proposed approach yields a benefit over relevant baselines. However, I do have several questions about the empirical validation.

Related to my point above, could the authors report experiments which illustrate the mechanism by which the proposed approach obtains improved bounds over Kallus & Zhou (2018a; 2021)? For instance, can the authors illustrate how the efficiency of the estimator yields tighter finite-sample bounds, and in turn, improves downstream policy learning? Or, similarly, that error in IPW weights propagates down to learned policies? Evidence along these dimensions would help the reader understand why the proposed method is necessary over Kallus & Zhou (2018a; 2021).

Further, it appears in several of the empirical results that the proposed method obtains high variance across runs. I find this surprising given that the estimation approach should in principle reduce variance in estimated policy value bounds. For example, confidence intervals are quite wide with Efficient + sharp as compared to baselines. We also see similar behavior in Figure 3. Could the authors explain why this is the case, and also include Kallus & Zhou (2018a; 2021) in Figure 3 for a clear comparison of variance across runs?

Additionally, while comparing regret against a fully randomized policy is a reasonable starting point, this seems overly simplified for a real-world experiment, especially because non-randomized baseline policies may yield more challenging distribution shift. Can the authors also report comparisons against other baseline policies?

Finally, given that the proposed estimator depends on learned nuisance functions, can the authors report details surrounding the procedure used to fit and select nuisance functions used to construct the doubly-robust estimates?

Overall, I see this work as providing a valuable contribution but do have significant concerns. I am open to re-considering my score if these concerns regarding significance and empirical validation are appropriately addressed.

**Questions:**

## Questions:
See questions raised above. Additionally:
- Kallus and Zhou also require "Strong overlap" to hold w.r.t. the true propensity - i.e., exists $ν > 0$ such that $e_a(x, u) ≥ ν, \; \forall a \in A$. Is such an assumption also needed here given the use of the MSM?
- Algorithm 2: Can cross-fitting be performed to improve sample efficiency?
- Figure 5: Why are other approaches not sensitive to the choice of sensitivity parameter? Can you show results for the full range, starting at $\Gamma=1$?

---

> ### Author Response · Authors · 2025-11-17
>
> Thank you very much for your helpful review of our paper!
>
>
> ### Responses to Weaknesses:
>
> **> W1: Comparison to Kallus & Zhou:**
>
> Thank you. Our contribution is not a conceptual departure from their marginal sensitivity model (MSM), but we offer a novel, non-trivial way to strengthen the estimation and optimization procedure under the MSM Below we provide a more pointed and technical comparison:
>
> **1.  Regret vs. value: sharp regret bounds do not imply sharp value bounds.**
>
> The method by Kallus & Zhou is explicitly designed to deliver **bounds on the regret** of a policy relative to a baseline, i.e. on the functional
>
> $$
> R(P) = V(\pi;P) - V(\pi_0;P), \qquad P \in \mathcal{P}(\Gamma).
> $$
>
> In case these bounds are sharp, it only guarantees that the interval
> $$
> [R_L, R_U] = [
> \inf_{P\in\mathcal{P}(\Gamma)} R(P),
> \sup_{P\in\mathcal{P}(\Gamma)} R(P)
> ]
> $$
> is the smallest valid identified set for the **regret**.
>
>
> However, this does **not** identify the sharp MSM bounds for the **value** of the policy $[V_L,  V_U]
> =$
>
> $$
> [
> \inf_{P\in\mathcal{P}(\Gamma)} V(\pi;P),
> \sup_{P\in\mathcal{P}(\Gamma)} V(\pi;P)
> ].
> $$
>
> The core issue is that the distributions maximizing regret and those maximizing
>
> baseline value **need not coincide**. Consequently, the value bounds implied by regret
>
> satisfy only the loose inequality
>
> $$
> \sup_{P\in\mathcal{P}(\Gamma)} V(\pi;P)
> \leq
> \sup_{P\in\mathcal{P}(\Gamma)}\\big(V(\pi;P)-V(\pi_0;P)\big)
> +
> \sup_{P\in\mathcal{P}(\Gamma)} V(\pi_0;P)
> $$
>
> and the inequality is **typically strict**. Thus, even a *sharp* regret interval  does **not** translate into *sharp* bounds on the policy value.
>
> In contrast, our estimator works directly with $(V(\pi)$ under the MSM and we provide a closed-form sharp identified set for the value itself. Hence, **our method is guaranteed to find the global optimal confounding-robust policy**.
>
> **2. How instability in IPW/Hájek weights propagates in Kallus & Zhou.**
>
> The method by Kallus & Zhou computes worst-case regret via a minimax IPW/Hájek objective over a set of feasible weights $U_\Gamma$. For each candidate policy, the inner optimization is a **linear–fractional program** over $U_\Gamma$, which is solved via a discrete threshold $k^\*$ (using ternary search) that determines which observations receive lower vs upper extreme weights.
>
> Because this worst-case construction depends on a **single global threshold**, estimation noise in the auxiliary variables $\lambda_k$, which themselves are inverse-propensity–weighted quantities, can cause the optimizer to select a suboptimal $k^*$. When this happens, a **local estimation error** in one $\lambda_k$ leads to a **global change** in the pattern of worst-case weights across **all observations**. Because of this, **IPW noise propagates into policy evaluation and regret**, as **small errors** in estimating $e(a \mid X)$ can alter the worst-case assignment **for all samples**. Hence, the estimation and optimization in Kallus & Zhou suffer from instability.
>
> **3. How our approach mitigates this instability.**
>
> Our estimator avoids this propagation mechanism entirely. Instead of relying on a discrete minimax search with IPW/Hájek ratios, we derive **closed-form sharp bounds on the value function**. Hence, one of our core contributions is to derive such closed-form sharp bounds.
>
> Further, we derive a **Neyman-orthogonal** estimator for our closed-form solution. Consequently, estimation errors in the nuisance functions do **not** propagate to our second-stage estimator of the value bound as first-order biases, but only at lower order. As a result, *our method can successfully circumvent the instabilities of Kallus & Zhou*.
>
> On top, we avoid any discrete thresholding step, so there is **no** propagation of single-point IPW errors to all samples, and no dependence on a global worst-case splitting index $k^*$. This yields substantially improved empirical performance.
>
> **4. Sharpness: differences from Kallus & Zhou.**
>
> Kallus & Zhou establish that their numerically solved regret interval is sharp for the marginal sensitivity model, but they do *not* derive sharp bounds on the value $V(\pi)$ itself. Their sharpness result concerns whether their **worst-case regret relative to a given baseline** is tight. In contrast, we derive **closed-form sharp bounds on the policy value**. These bounds apply to each policy $\pi$ individually and are not relative to a baseline policy. Our results therefore extend the notion of sharpness in a novel way: we obtain the **globally sharp identified set for $V(\pi)$** itself, not only for regret differences.

---

> ### Author Response · Authors · 2025-11-17
>
> **5. Semiparametrically efficient estimator.**
>
> Finally, our one-step estimator is derived from the **efficient influence function (EIF)** of the value bound. This provides the standard semiparametric guarantees (optimal asymptotic variance & insensitivity to first-stage errors), which the IPW/Hájek estimator used by Kallus & Zhou does **not** satisfy. Hence, our method has several theoretical benefits compared to Kallus & Zhou as a result of semiparametric efficiency.
>
> **Action:** We have added a new **Supplement B,** where we provide additional background for the baseline method, and where we clarify these distinctions: We provide (i) closed-form sharp value bounds, (ii) a fully EIF-based efficient estimator, and (iii) improved numerical stability due to the absence of discrete minimax search and global weight thresholding.
>
> **> W2: Relation to Rambachan et al. and MOSM:**
>
> Thank you for highlighting this work!
>
> Their work addresses a different problem: a *binary* selection decision $D\in{0,1}$ with a ***binary*** selectively observed outcome $Y^\ast$, and they derive sharp bounds on *prediction and fairness* functionals (risk, ROC, precision–recall, etc.) under a **Mean Outcome Sensitivity Model (MOSM)** that directly bounds the outcome gap $\delta(x) = P(Y^\ast=1\mid D=0,X=x) - P(Y^\ast=1\mid D=1,X=x)$. Their sharpness guarantees are always with respect **to this MOSM**.
>
> By contrast, we study **offline policy learning** under a **Marginal Sensitivity Model (MSM) on treatment assignment** with (possibly) multi-valued actions $A\in\mathcal{A}$ and generally **non-binary**, real-valued outcomes $Y$. Our estimand is the **policy value** ($V(\pi)$) for a stochastic policy $\pi(a\mid x)$. We derive **closed-form, sharp bounds** on $V(\pi)$ under the MSM and construct a semiparametrically efficient EIF-based estimator for these bounds.
>
> Importantly, Rambachan et al. themself argue that **propensity-based models such as the MSM imply a particular special case of the MOSM** (via observed-outcome bounds obtained from Bayes’ rule). Hence, the implication is **MSM** $\Rightarrow$ **a specific MOSM**, **not** the other way around, and they explicitly note that these induced outcome bounds “may not exhaust all implications” of the original sensitivity model. Thus, **their bounds are sharp for the outcome-based MOSM they posit, but not necessarily sharp for the underlying MSM**. In contrast, our analysis works **directly at the level of the MSM** for general finite $\mathcal{A}$, real-valued outcomes, and characterizes the **sharp MSM-identified set for the policy value** $V(\pi)$. Put simply, only our bounds are guaranteed to be sharp in our setting.
>
> In sum, (1) our estimation task is entirely different, (2) our employed sensitivity model is different, and (3) our action and outcome spaces are more general.
>
> **Action:** We added the above work to our extended related work section in ou **revised Supplement A** with a discussion on key differences.

---

> ### Author Response · Authors · 2025-11-17
>
> **> W3: Empirical validation and differences to the baseline**
>
> Thank you. Our method offers several advantages over the method by Kallus & Zhou:
>
> **Key differences:**
>
> (i) We provide sharp bounds on the value function and, hence, guarantee that we optimize for the optimal confounding robust policy (Proposition 4.1 and Theorem 4.4). In contrast, Kallus & Zhou only search for an improvement over a baseline policy (e.g., a randomized policy), which does **not** guarantee finding the optimal policy (please see our response to **W1**).
>
> (ii) Our bounds are sharp, and we deliver a closed-form expression, whereas Kallus & Zhou rely on a ternary search algorithm. In their algorithm, **IPW noise propagates into policy evaluation and regret**, as **small errors** in estimating $e(a \mid X)$ can change the worst-case assignment **for all samples**. Put differently, one bad estimate can mess up the whole batch, which makes their algorithm unstable.
>
> (iii) Our estimator is **semiparametrically efficient**, which means that it *achieves the lowest possible asymptotic variance among all unbiased estimators*. Specifically, our estimator is Neyman-orthogonal, which means that estimation errors in the nuisance functions do **not** propagate as first-order bias to the final stage estimate. In contrast, Kallus & Zhou use an IPW-type Hajek-estimator, which can lead to **instabilities** when estimated propensities are extreme (close to 0 or 1), and has **finite sample bias**.
>
> **Empirical validation: Propensity scores**
>
> Regarding empirical validation, we are happy to report **new experimental results** (see our **new Supplement C)**, where we highlight how estimation errors in more complex propensity scores propagate in both methods. Thank you  for this idea – as you will see below, it nicely validates the above theoretical insights and confirms the theoretical benefits of our method.
>
> For this, we extend our synthetic data generating process from before: we do not use binary treatments, but increase the number $d_a$ of available treatments $a \in \{0,1,...,d_a-1\}$. We fix the ground truth unobserved confounding strength to $\Gamma^*=5$ in the data, and for both methods correctly specify the sensitivity parameter as $\Gamma=5$.
>
> In this setup, correctly estimating the propensity score becomes more complex (put simply, multi-class classification is harder than binary classification). By increasing $d_a$, we can therefore study how well our method and the method by Kallus & Zhou react under an increasingly more difficult estimation task.
>
> The empirical results confirm our theoretical prediction:
>
> - **Our Efficient + Sharp estimator remains stable** and even **improves** as $d_a$ increases.
>
> The closed-form value bounds and the EIF-based one-step correction prevent first-order propagation of nuisance errors. Even when the multiclass propensities become harder to estimate, the orthogonality and sharpness of our estimator ensure a stable, confounding-robust policy.
>
> - **Kallus & Zhou deteriorate as $d_a$ increases.**
>
> Their procedure again appears to revert to the baseline policy (as in our real-world data experiments). The reason is that their worst-case regret computation relies on ordering the auxiliary IPW scores $\lambda_i(\pi,\pi_0)$ and selecting a single global threshold $k^\*$. When multiclass propensities are noisy, the $\lambda_i$’s become extremely variable, which makes the argmax over feasible weights
>
> $$
>   \sup_{w\in U_\Gamma} \frac{\sum_i w_i\,\lambda_i}{\sum_i w_i}
> $$
>
> flip between many local optima. In practice, the ternary search tends to settle on a $k^\*$ that assigns mostly *lower* weights to all non-baseline units. This produces an overly conservative worst-case regret estimate, and makes every non-baseline policy look unsafe and pushes the algorithm toward the baseline policy for all runs.
>
> In short:
>
> - **Our method**: nuisance errors *do not* propagate as first-order bias -> bounds remain tight -> policy learning improves with more actions.
>
> - **Kallus & Zhou**: nuisance errors *do* propagate as first-order biases directly through the global thresholding step -> regret becomes uniformly pessimistic -> the policy collapses to the baseline.
>
> **Action:** We add **new experimental results** with increasingly more complex propensity scores in our **new Supplement C**. Therein, we explain why our method performs well, and why the baseline performance deteriorates.

---

> ### Author Response · Authors · 2025-11-17
>
> **> W4: Variance**
>
> Thank you for pointing this out. Upon reading your comment, we realized that there may be confusion regarding the difference between (i) variance of the estimator, and (ii) variance of the objective. The latter is what drives the differences in variance:
>
> Semiparametric efficiency (as in our method) guarantees minimal variance for estimating   $V(\pi)$ **for a fixed policy**.
>
> However, the variance reported in Figures 3–5 reflects **variability in the learned policy**,  not inefficiency of the value estimator.
>
> To be more precise:
>
> Our method optimizes **sharp bounds on** $V(\pi)$, and these bounds remain sensitive to
>
> differences between policies across the entire range of $\Gamma$. As a consequence, the  policy selected by our method varies more across repetitions, which  induces higher variance in the *policy-level performance plots*.
>
> In contrast, the Kallus & Zhou method optimizes **worst-case regret relative to a fixed baseline**. For increasing $\Gamma$, the regret objective becomes extremely flat, so that all non-baseline policies look equally bad under worst-case confounding, and the optimizer collapses to (or very close to) the baseline policy.  Thus, the learned policy becomes almost constant across runs, which yields *artificially low variance* in their performance curves. The lower variance is a consequence of the **insensitivity of their worst-case regret objective**, not superior estimator efficiency. The resulting *policy* has low variability because the algorithm is **conservative**, not because the estimator has low variance.
>
> Following your request, we added the baseline by Kallus & Zhou to our experiments in Figure 3, and the results confirm the above: for mis-specified, low $\Gamma\leq \Gamma^*$, their method has good performance, which, however, quickly deteriorates as we increase the sensitivity parameter. This is expected: the Kallus & Zhou estimator is too *conservative*: for small $\Gamma$, its bounds are fairly tight, and worst-case regret still sometimes favors non-baseline policies. But once $\Gamma$ becomes moderately large, the method quickly reverts to the baseline policy (see **Q3** for a more detailed discussion). Since the true DGP uses $\Gamma = 7$, their method is effectively analyzing the data  as if it were *less confounded* than it actually is. Underestimating $\Gamma$ can temporarily yield good empirical performance, but this is **not** due to robustness. Instead, it is an artifact of optimistic modeling. As we increase $\Gamma$, their objective collapses, while our sharp bounds correctly reflect the uncertainty under the MSM.
>
> **Action:** We **added the baseline** to our results in Figure 3. Therein, we further explain the above differences and **clarify the variance properties**. Further, we provide a new **Supplement B** highlighting the **technical differences** of our method from Kallus & Zhou.

---

> ### Author Response · Authors · 2025-11-17
>
> **> W5: Comparison to randomized policy**
>
> Thank you for this suggestion. We would like to clarify the rationale behind choosing a fully randomized policy as the baseline in our regret comparisons.
>
> Our goal is *not* to demonstrate improvement over a policy that would be optimal under unconfoundedness.  Instead, the setting we study is one where **unobserved confounding is present**, and therefore *no baseline policy that assumes unconfoundedness is well-defined or interpretable*. In such environments, any non-randomized baseline would implicitly encode structural assumptions that are violated, making its “regret’’ difficult to interpret and potentially misleading.
>
> The randomized policy plays a special role: it is the **only** policy whose value remains identified under arbitrary unobserved confounding, and therefore serves as a **neutral “no-information’’ reference point**. Regret relative to this RCT baseline measures exactly what we intend:
>
>  *How much a policy learner can improve over doing nothing when unobserved confounding prevents identification of the optimal policy.*
>
> This is also aligned with our real-world RCT experiment. The original dataset is unconfounded since it is an RCT; we inject controlled levels of unobserved confounding to create a realistic sensitivity-analysis environment. In such a setting, the randomized policy is again the correct baseline, because **no other non-randomized policy can be justified** once confounding has been introduced.
>
> Nonetheless, to address the your comment, we now include **additional results on a second real-world dataset (IHDP**) where we introduce *lower* levels of unobserved confounding. Across all these settings, we consistently observe that **our method yields the largest improvement** over the randomized baseline. Our results further reinforce the intended message: when confounding is present, the meaningful question is not whether we outperform an unconfounded “oracle’’ policy, but how much gain we obtain over the only robust reference point, which is the uninformed RCT baseline.
>
> **Action:** We clarify the intention behind our RCT baseline that we seek to improve upon. Further, we report **new experimental results on the IHDP dataset** in **Supplement D.**
>
>
> **> W6: Estimation of nuisance components**
>
> Thank you. In our **Supplement H**, we report implementation details of all methods and their nuisance estimators. We are happy to provide more details if anything remains unclear.

---

> ### Author Response · Authors · 2025-11-17
>
> ### Responses to Questions:
>
> **> Q1: Strong overlap**
>
> Thank you for this interesting question! We double checked our assumptions and arrived at the following conclusion:
>
> We do indeed need **strong overlap for the observed propensity** $e(a\mid X=x)$, i.e. it must be bounded away from 0 (and 1) on the support of X. This is what guarantees that the EIF terms involving $1/e(a\mid X)$ have finite variance and that the one-step estimator has root-(n) asymptotics.
>
> However, we do **not** separately assume strong overlap for the latent $e(a\mid X,U)$; the MSM restriction with finite $\Gamma$ plus strong overlap on (e(a\mid X)) is enough to make the bounds and EIF well-behaved.
>
> **Action:** We **updated** the (ii) Positivity assumption in our paper accordingly. Thank you for pointing this out!
>
> **> Q2: Cross-fitting**
>
> Thank you. Indeed, we can perform cross-fitting: the simple sample split in Algorithm 2 can be replaced by standard $K$-fold cross-fitting of the nuisance estimators (i.e., fit $\hat{\eta}$ on $K-1$ folds, evaluate the efficient estimator of the bound of the policy value on the held-out fold, aggregate over folds).
>
> Thereby, we use every observation both for nuisance estimation and for policy learning, and improve small-sample efficiency. Importantly, this remains compatible with Neyman-orthogonality of the EIF; for simplicity and computational reasons we implemented a single split in the current version.
>
> **Action:** We now highlight in our **revised Algorithm 1** that cross-fitting can be performed to increase sample efficiency.
>
> **> Q3: Baseline performance Fig. 5**
>
> Thank you.
>
> For DR and IPW, the insensitivity occurs since they *do not use* the sensitivity parameter at all. They assume unconfoundedness and optimize an IPW/DR objective with nominal propensities, so both the learned policies and their value functions are *independent* of $\Gamma$. Their flat lines are thus expected.
>
> Regarding the baseline by Kallus and Zhou, we have the following intuition for their insensitivity to $\Gamma$. There are two different things going on:
>
> We optimize a different functional compared to Kallus:
>
> **Our method:**
>
> For each policy $\pi$ and sensitivity parameter $\Gamma$, we have a **closed-form sharp upper bound** $ V^{+,\*}(\pi;\Gamma)$. We choose
>  $$ \arg\max_{\pi\in\Pi} V^{+,*}(\pi;\Gamma)$$
>
> The bound is a **monotone** function of $\Gamma$. As $\Gamma$ changes, the upper bounds for different policies move at different rates, so **the ordering between policies can change frequently**. Hence, our learned optimal policy and its policy value can react strongly to $\Gamma$, as we see in Fig. 5.
>
>
>
> **Kallus’ method:**
>
> They don’t optimize a bound on the value itself, but **worst-case regret vs an arbitrary baseline** $\pi_0$:
>
>  $$R_{\pi_0}(\pi;\Gamma) = \sup_{W \in U_\Gamma} \text{(Hájek/IPW regret with weights }W),$$
>
>  and pick
>
> $$\arg\min_{\pi\in\Pi} R_{\pi_0}(\pi;\Gamma).$$
>
> Once $\Gamma$ is large enough that the worst-case regret of any non-baseline policy looks bad, the optimizer just sticks to (or very close to) the baseline policy $\pi_0$. In that regime, increasing $\Gamma$ changes the value of the objective but does not change the minimizer, so the curve is flat.
>
> In short, the behaviour by Kallus’ baseline is most likely due to the fact that they do not optimize w.r.t. the globally optimal confounding-robust baseline, but only for an improvement over a baseline policy. We kindly refer to our answer **W1 on** why this is generally not optimal.
>
> On top of that, Kallus’ method relies on a discrete optimization (ternary search over a threshold $k^\*$). For and a fixed policy, the inner supremum over $U_\Gamma$ can be written as a maximum over $k \in \{1,\dots,n\}$, and for wide ranges of $\Gamma$ the same index $k^\*(\Gamma)$ remains optimal. In these ranges, the identity which observations receive upper versus lower feasible weights **remains unchanged**, and changes in $\Gamma$ **only rescale** the same configuration. As a result, the policy $\arg\min_\pi R_{\pi_0}(\pi;\Gamma)$ tends to be **piecewise constant** in $\Gamma$, which is exactly what we observe in our experiments.
>
> **Smaller values of $\Gamma$:**
>
> Finally, we are happy to report results for $\Gamma<6$. We can see that, for values close to 1, the naive learners improve upon our method. When we choose $\Gamma$ small, our method does **not appropriately guard against confounding**, while it still requires estimation of more complex nuisance functions (conditional quantiles, truncated moments, etc.) than, for example, standard DR/IPW. In other words, it pays the variance cost of a full sensitivity model but receives none of its robustness benefits. Hence, the results are **expected**.
>
> **Action:** We **add the results** for small values of $\Gamma$ to our experiments. Therein, we explain when and why our method performs best. Further, we add a new **Supplement B** where we discuss the method by Kallus and Zhou in more detail.

---

> ### Author Response · Authors · 2025-11-28
>
> Dear reviewer d6rY,
>
> We thank you for your helpful and positive feedback on our submission.
>
> We hope that our rebuttal could clarify your remaining question. Otherwise, we are happy to provide further insights!
>
> If you feel comfortable with it, we would highly appreciate it if you would consider increasing your score.

---

### Official Review · Reviewer_emv2 · 2025-11-04

**Soundness:** 3
**Presentation:** 3
**Contribution:** 2
**Rating:** 6
**Confidence:** 4

**Summary:**

This paper considers the problem of unobserved confounding in offline policy learning. They assume that the unobserved confounding satisfies the marginal sensitivity model (Tan, 2006), which is often used in the sensitivity analysis literature (Aronow and Lee 2013, Miratrix et al. 2018, Zhao et al. 2019, Yadlowsky et al. 2018, Kallus et al. 2018, Kallus and Zhou 2020).

**Strengths:**

- The paper is clearly written and easy to understand.
- The main contribution of this paper is a semiparametrically efficient estimator for offline robust policy learning problem, arguing that the approach of Kallus and Zhou 2020 may be unstable due to the dependence on inverse propensity weights. Instability of inverse propensity weights is a known problem that can lead to instability of estimators.
- They propose a naive plug-in estimator for the optimal robust policy but note that it will suffer from first-order bias. Then, they derive the semiparametrically efficient estimator that does not suffer from the first-order bias from the estimation of nuisance components
- The theoretical contributions are sound.

**Weaknesses:**

- The problem of robust offline policy learning under the marginal sensitivity model and Rosenbaum selection model is quite well-studied, e.g. Aronow and Lee 2013, Miratrix et al. 2018, Zhao et al. 2019, Yadlowsky et al. 2018, Kallus et al. 2018, Kallus and Zhou 2020. Furthermore, other works such as Bruns-Smith and Zhou, 2023 consider dynamic policy learning. So, the problem that the authors aim to solve has limited novelty. Nevertheless, this paper does cite and reference many of the relevant works in this area and I believe they do make a technical contribution (in terms of semi-parametric efficiency of their estimator), relative to the Kallus and Zhou 2020.
- The new estimator appears to only provide modest improvements over the naive plug-in estimator.

**Questions:**

N/A

---

> ### Author Response · Authors · 2025-11-17
>
> Thank you very much for your positive review of our paper!
>
>
> ### Responses to Weaknesses:
>
> **>W1: Related work**
>
> Thank you for pointing to these interesting works! There are, however, important differences to our work:
>
> **Average treatment effect under sensitivity models:**
>
>
>
> * Aronow and Lee (2013) and Miratrix et al. (2018) study partial identification of the population mean. For this, they work with Hajek-type estimators (which have **finite-sample bias**) and solve linear-fractional optimization problems to obtain bounds on the mean. Their focus is on estimating a population mean, and **not** on policy value or regret. Further their estimators are **not** semiparametrically efficient. In contrast, we focus on the policy value / regret and our method is semiparametrically efficient.
> * Zhao et al. (2019) work under the marginal sensitivity model and develop asymptotic confidence intervals for IPW-type estimators of the mean response, for which they rely on bootstrap inference. In contrast, our work is not on the mean response but on the policy value / regret and we derive sharp bounds in closed form.
> * Yadlowsky et al. (2018) construct estimators for the corresponding **ATE bounds**. However, their setting is limited to **binary treatment**, and their target is the treatment effect, and **not** an optimal policy value bound over a potentially multi-action policy class. In contrast, our method is applicable to discrete treatments, our target estimand is the policy value / regret (rather than treatment effect bounds).
>
> By contrast, our work is explicitly about **confounding-robust offline policy learning** for **a discrete action space**. Ours is the only work toto derive *sharp bounds on the value function and regret of arbitrary policies* under the marginal sensitivity model, and we give a closed-form expression for the sharp value bounds together with a **semiparametrically efficient, Neyman-orthogonal one-step estimator** that is tailored to policy learning (i.e., can be combined with complex policy classes without re-solving a difficult optimization problem for each policy).
>
> **Dynamic / offline RL under unobserved confounding:**
>
>
>
> * Bruns-Smith and Zhou (2023) consider a **different setting**: dynamic policies in Markov decision processes under unobserved confounding. Their estimator has to deal with sequences of states and actions and therefore estimates **ratios of transition probabilities**:
>
> Hence, the partial identification bounds and estimators are **entirely differen**t from the static, one-step contextual bandit setting we study, and cannot be dropped into our setting without substantial modification. Further, they show via a counterexample that their robust FQE/FQI method can yield **non-sharp** value bounds when there are more than two actions, whereas in our setting we derive **closed-form sharp bounds** for arbitrary finite action spaces together with an efficient influence-function-based estimator.
>
>
> **Kallus and Zhou:**
>
> Our contribution focuses on the same setting as Kallus and Zhou. To this end, we focus on the same marginal sensitivity model as in their work, but we make *novel, non-trivial technical contributions to improve the estimation and optimization under the MSM*. Below we provide a more pointed and technical comparison.
>
> **1.  Regret vs. value: sharp regret bounds do not imply sharp value bounds.**
>
> The method by Kallus & Zhou is explicitly designed to deliver **bounds on the regret** of a policy relative to a baseline, i.e. on the functional
>
> $$
> R(P) = V(\pi;P) - V(\pi_0;P), \qquad P \in \mathcal{P}(\Gamma).
> $$
>
> In case these bounds are sharp, it only guarantees that the interval
> $$
> [R_L, R_U] = [
> \inf_{P\in\mathcal{P}(\Gamma)} R(P),
> \sup_{P\in\mathcal{P}(\Gamma)} R(P)
> ]
> $$
> is the smallest valid identified set for the **regret**.
>
>
> However, this does **not** identify the sharp MSM bounds for the **value** of the policy $[V_L, V_U]
> =$
>
> $$
> [
> \inf_{P\in\mathcal{P}(\Gamma)} V(\pi;P),
> \sup_{P\in\mathcal{P}(\Gamma)} V(\pi;P)
> ].
> $$
>
> The core issue is that the distributions maximizing regret and those maximizing
>
> baseline value **need not coincide**. Consequently, the value bounds implied by regret
>
> satisfy only the loose inequality
>
> $$
> \sup_{P\in\mathcal{P}(\Gamma)} V(\pi;P)
> \leq
> \sup_{P\in\mathcal{P}(\Gamma)}\\big(V(\pi;P)-V(\pi_0;P)\big)
> +
> \sup_{P\in\mathcal{P}(\Gamma)} V(\pi_0;P)
> $$
>
> and the inequality is **typically strict**. Thus, even a *sharp* regret interval  does **not** translate into *sharp* bounds on the policy value.
>
> In contrast, our estimator works directly with $(V(\pi)$ under the MSM and we provide a closed-form sharp identified set for the value itself. Hence, **our method is guaranteed to find the global optimal confounding-robust policy**.

---

> ### Author Response · Authors · 2025-11-17
>
> **2. How instability in IPW/Hájek weights propagates in Kallus & Zhou.**
>
> The method by Kallus & Zhou computes worst-case regret via a minimax IPW/Hájek objective over a set of feasible weights $U_\Gamma$. For each candidate policy, the inner optimization is a **linear–fractional program** over $U_\Gamma$, which is solved via a discrete threshold $k^\*$ (using ternary search) that determines which observations receive lower vs upper extreme weights.
>
> Because this worst-case construction depends on a **single global threshold**, estimation noise in the auxiliary variables $\lambda_k$, which themselves are inverse-propensity–weighted quantities, can cause the optimizer to select a suboptimal $k^*$. When this happens, a **local estimation error** in one $\lambda_k$ leads to a **global change** in the pattern of worst-case weights across **all observations**. Because of this, **IPW noise propagates into policy evaluation and regret**, as **small errors** in estimating $e(a \mid X)$ can alter the worst-case assignment **for all samples**. Hence, the estimation and optimization in Kallus & Zhou suffer from instability.
>
> **3. How our approach mitigates this instability.**
>
> Our estimator avoids this propagation mechanism entirely. Instead of relying on a discrete minimax search with IPW/Hájek ratios, we derive **closed-form sharp bounds on the value function**. Hence, one of our core contributions is to derive such closed-form sharp bounds.
>
> Further, we derive a **Neyman-orthogonal** estimator for our closed-form solution. Consequently, estimation errors in the nuisance functions do **not** propagate to our second-stage estimator of the value bound as first-order biases, but only at lower order. As a result, *our method can successfully circumvent the instabilities of Kallus & Zhou*.
>
> On top, we avoid any discrete thresholding step, so there is **no** propagation of single-point IPW errors to all samples, and no dependence on a global worst-case splitting index $k^*$. This yields substantially improved empirical performance.
>
> **4. Sharpness: differences from Kallus & Zhou.**
>
> Kallus & Zhou establish that their numerically solved regret interval is sharp for the marginal sensitivity model, but they do *not* derive sharp bounds on the value $V(\pi)$ itself. Their sharpness result concerns whether their **worst-case regret relative to a given baseline** is tight. In contrast, we derive **closed-form sharp bounds on the policy value**. These bounds apply to each policy $\pi$ individually and are not relative to a baseline policy. Our results therefore extend the notion of sharpness in a novel way: we obtain the **globally sharp identified set for $V(\pi)$** itself, not only for regret differences.
>
> **5. Semiparametrically efficient estimator.**
>
> Finally, our one-step estimator is derived from the **efficient influence function (EIF)** of the value bound. This provides the standard semiparametric guarantees (optimal asymptotic variance & insensitivity to first-stage errors), which the IPW/Hájek estimator used by Kallus & Zhou does **not** satisfy. Hence, our method has several theoretical benefits compared to Kallus & Zhou as a result of semiparametric efficiency.
>
> **Action:** We **added the above literature** to our related work section. Further, we provide a **new Supplement B**, where we explain the method by Kallus & Zhou in more detail, and point to their limitations.
>
> **>W2: Improvement over plug-in estimator**
>
> Thank you for your feedback. Please note that, in **Figure 4,** the **relative improvement of our method over a plug-in estimator of the sharp bounds is up to ~50%**, which demonstrates our optimal asymptotic variance properties.

---

> ### Author Response · Authors · 2025-11-28
>
> Dear reviewer emv2,
>
> We thank you for your helpful and positive feedback on our submission.
>
> We hope that our rebuttal could clarify your remaining question. Otherwise, we are happy to provide further insights!
>
> If you feel comfortable with it, we would highly appreciate it if you would consider increasing your score.

---

### Author Response · Authors · 2025-11-17
**Response to all reviewers**

We thank the reviewers for the comprehensive and helpful feedback on our work.

To improve our work, we added the following improvements to our revised paper (highlighted in $\color{blue}{\text{blue}}$):



* **More discussions:** We added a **new Supplement B**, where we provide a more technical discussion on the key differences of our method from the baseline by Kallus & Zhou.
* **New sensitivity experiment:** We added a **new experiment** in our **new Supplement C**, where we study the sensitivity of our method and the baseline with respect to **more complex treatment propensities**. For this, we use synthetic data and increase the action space. As expected, our method performs well, whereas the baseline deteriorates.
* **New experiment on IHDP:** We added a **new experiment** in our **new Supplement D** on another dataset, the **IHDP data**. Therein, we introduce unobserved confounding and study the performance of our method against all baselines. Our results **confirm** our previous findings: our method finds the best-performing, confounding-robust policy.
* **More references:** We **added more references** as suggested by the reviewers to our related work in our **revised Section 2,** and to the extended related work in **Supplement A**.
* **Clarifications:** We **revised our technical assumptions** (strong overlap, standard assumptions for pathwise differentiability), and now clearly state that our improvement guarantees in Theorem 4.4 **simultaneously** hold for all policies $\pi \in \Pi$.
* We highlight the connection between the EIF, the canonical gradient, and semi-parametric efficient estimators in our proofs.
* We made some **smaller additions** to the paper (e.g., we explained the poor baseline performance in the real-world dataset, we highlighted cross-fitting to improve sample-efficiency in Algorithm 1, and we clarified  the asymptotic behaviour of neural networks as a parametric policy class).

We are confident that, with the help of the reviews and our improvements, our work will be an important contribution to the community. Finally, we wanted to say “thank you” – we truly appreciate the help, which allowed us to revise our work and make important improvements.

---

### Meta-Review · Area_Chair_njeq · 2025-12-24

**Summary:**

**Summary of the contributions** : The paper proposes a robust policy estimation under unobserved confounding. Typically, under the MSM (marginal sensitivity model) i.e., capturing the deviation of the observed propensity and nominal propensity score, this problem is typically solved as a min-max optimization. Instead this work focuses on deriving sharp bounds on the value function reducing the min-max problem to a minimization problem. In addition the paper proposes a semi-parametrically efficient estimator for this bound. Empirical evaluation in synthetic data, and the stroke trial data demonstrates improvement over typical baselines such as kallus & zhou 2018 who propose the min-max optimization.

** Reviewer comments/concerns**:
1. The technical contribution is limited but novel, especially in the context of Kallus and Zhou 2020.
2. Empirical improvement is modest but significant
3. Reviewers asked for additional contextualization compared to Kallus & Zhou 2018, and Rambachan, Coston, and Kennedy 2022, who derive sharp bounds on the marginal outcome sensitivity model.
4. Additional empirical ablations on dissecting the source of benefits of using the bounds proposed in the paper, especially compared to min-max frameworks, in terms of the associated bias-variance tradeoff, the general mechanism of statistical gains, and impact of choice of nuisance designs.
5. Completeness of proofs, and specificying conditions,
6. Reviewer points out that Theorem 4.4. may need a uniform bound as opposed something pointwise in $\pi$

**Reviewer Concerns:**

During the rebuttal, several clarifications have been added that seem to have strengthen the manuscript.

First, additional context on prior work has been added. Particularly, new empirical analysis on failure modes of the Kallus & Zhou, 2018 estimator compared to the paper describing how the proposed estimator is robust to increasing number of arms by stabilizing nuisance estimation in the semiparametric estimation of the bounds on the value function.

In addition appendix includes detailed description of the distinction of the Hajek estimator of Kallus & Zhou (obtaining sharp bound on the regret) vs the current paper (emphasizing a sharp bound on the value function). Finally, the authors have also added additional empirical results on the IHDP dataset.

The rebuttal itself includes, in my opinion, sufficient responses to concerns of emv2 but who has not engaged in responses.

Based on reviewer feedback the algorithm has been improved to incorporate cross-fitting, and the proof is updated to include the uniform bound.

**Reviewer Scores:**

I think almost all reviewer concerns are addressed, though all reviewers did not engage with the rebuttal. Considering the author responses and manuscript updates, I am recommending an accept.

---

### Decision · Program_Chairs · 2026-01-26

Accept (Poster)